# BONSAI: GRADIENT-FREE GRAPH CONDENSATION FOR NODE CLASSIFICATION

**Mridul Gupta**[1*]  **Samyak Jain**[2*]  **Vansh Ramani**[2]  **Hariprasad Kodamana**[1,3,4]  **Sayan Ranu**[1,2]

[1]Yardi School of Artificial Intelligence      [2]Department of Computer Science
[3]Department of Chemical Engineering

Indian Institute of Technology Delhi, New Delhi, 110016, India
[4]Indian Institute of Technology Delhi, Abu Dhabi, Zayed City, Abu Dhabi, UAE

{mridul.gupta@scai,cs5200667@,cs5230804@,kodamana@,sayanranu@cse}.iitd.ac.in

## ABSTRACT

Graph condensation has emerged as a promising avenue to enable scalable training of GNNs by compressing the training dataset while preserving essential graph characteristics. Our study uncovers significant shortcomings in current graph condensation techniques. First, the majority of the algorithms paradoxically require training on the full dataset to perform condensation. Second, due to their gradient-emulating approach, these methods require fresh condensation for any change in hyper-parameters or GNN architecture, limiting their flexibility and reusability. To address these challenges, we present BONSAI, a novel graph condensation method empowered by the observation that *computation trees* form the fundamental processing units of message-passing GNNs. BONSAI condenses datasets by encoding a careful selection of *exemplar* trees that maximize the representation of all computation trees in the training set. This unique approach imparts BONSAI as the first linear-time, model-agnostic graph condensation algorithm for node classification that outperforms existing baselines across 7 real-world datasets on accuracy, while being 22 times faster on average. BONSAI is grounded in rigorous mathematical guarantees on the adopted approximation strategies, making it robust to GNN architectures, datasets, and parameters.

## 1 INTRODUCTION AND RELATED WORKS

Graph Neural Networks (GNNs) have shown remarkable success in various predictive tasks on graph data such as node classification and link prediction (Veličković et al., 2018; Kipf & Welling, 2017; Hamilton et al., 2017; Nishad et al., 2021), modeling physical systems (Bhattoo et al., 2022; Thangamuthu et al., 2022; Bishnoi et al., 2023; 2024), and spatio-temporal data (Jain et al., 2021; Gupta et al., 2023). However, real-world graphs often contain millions of nodes and edges making the training pipeline slow and computationally demanding (Hu et al., 2021). This limitation hinders their adoption in resource-constrained environments (Miao et al., 2021) and applications dealing with massive datasets (Hu et al., 2021). *Graph condensation* (also called graph distillation) has emerged as a promising way to bypass this bottleneck (Jin et al., 2021; Liu et al., 2024a;b; Zheng et al., 2023). The objective in graph condensation is to synthesize a significantly smaller condensed data set that retains the essential information of the original data. By training GNNs on these condensed datasets, we can achieve comparable performance while reducing computational overhead and storage costs. This makes GNNs more accessible and practical for a wider range of applications, including those with limited computational resources and large-scale datasets. In this work, we study the problem of graph condensation for node classification.

### 1.1 EXISTING WORKS AND THEIR LIMITATIONS

Table 1 presents existing graph condensation algorithms proposed for node classification and their characterization across various dimensions. We omit listing DosCond (Jin et al., 2022), MiRAGE (Gupta et al., 2024), and KiDD (Xu et al., 2023) in Table 1 since they are designed for graph classification, whereas we focus on node classification.

---

*Denotes equal contribution.

Table 1: Characterization of existing graph condensation algorithms. Cells shaded in Red indicate the presence of an undesirable property, while Green represents their absence.

| Algorithm | Requires training GNN on full-dataset | Condenses to a fully-connected graph[1] | Model-specific condensation |
|---|---|---|---|
| GCOND (Jin et al., 2021) | ✓ | ✓ | ✓ |
| SGDD (Yang et al., 2023) | ✓ | ✓ | ✓ |
| SFGC (Zheng et al., 2023) | ✓ | ✓ | ✓ |
| GC-SNTK (Wang et al., 2024) | ✓ | ✗ | ✗ |
| EXGC (Fang et al., 2024) | ✓ | ✓ | ✓ |
| GEOM (Zhang et al., 2024) | ✓ | ✗ | ✓ |
| GDEM (Liu et al., 2024a) | ✗ | ✓ | ✗ |
| GCSR (Liu et al., 2024b) | ✓ | ✓ | ✓ |
| BONSAI | ✗ | ✗ | ✗ |

● **Full GNN training is a pre-requisite:** The fundamental requirement of data condensation is that it should require less computational resources and time than training on the full dataset. However, this basic premise is violated by majority of the techniques (See Table 1) since they adopt a design where training the target GNN on the full trainset is a prerequisite to condensation. These algorithms adopt a gradient-dependent optimization framework. They first train a GNN on the full trainset and extract gradients of model parameters over epochs. The condensation process is formulated as an optimization problem to create a condensed dataset that replicates the gradient trajectory observed in the original training set. This need for full dataset training for condensation contradicts its fundamental premise, which we address in our work.

● **Node compression vs. Edge complexity**: In a message-passing GNN, the computation cost of each forward pass is $\mathcal{O}(|\mathcal{E}|)$, where $\mathcal{E}$ denotes the set of edges. Consequently, the computational effectiveness of graph condensation is primarily determined by the reduction in edge count between the original and condensed graphs, rather than node count alone. However, current graph condensation algorithms (see Table 1) quantify the condensation ratio based on the node count. This creates a fundamental disconnect between the metric used to evaluate compression (node count) and the computational goal of reducing $\mathcal{O}(|\mathcal{E}|)$, which directly governs training and inference time. Consequently, this misalignment can lead to a condensed graph that is less efficient than the original.

● **Model-specific condensation:** Gradients of model weights are influenced by the specific GNN architecture and hyper-parameters (e.g., number of layers, hidden dimensions, etc.). Consequently, any architectural change, such as switching from a GCN (Kipf & Welling, 2017) to a GAT (Veličković et al., 2018), or hyper-parameter adjustments, necessitates a new round of condensation.

## 1.2 CONTRIBUTIONS

To address the limitations outlined in Table 1, we present BONSAI[2].

- **Gradient-free condensation:** Instead of replicating the gradient trajectory, BONSAI emulates the distribution of input data processed by message-passing GNNs. By shifting the computation task to the *pre-learning phase*, BONSAI achieves independence from hyper-parameters and model architectures as long as it adheres to a message-passing GNN framework like GAT, GCN, GRAPHSAGE, GIN, etc. Moreover, this addresses a critical limitation of existing graph condensation algorithms that necessitates training on the entire training dataset.
- **Novel algorithm design:** BONSAI is empowered by the observation that any message-passing GNN decomposes a graph of $n$ nodes into $n$ rooted *computation trees*. Furthermore, topologically similar computation trees generate similar embeddings regardless of the GNN being used (Xu et al., 2019; Togninalli et al., 2019). BONSAI exploits this observation to identify a small subset of diverse computation trees, called *exemplars*, that are located in dense regions and thereby representative of the full set. Hence, the induced subgraph spanned by the exemplars forms an effective condensed set.
- **Empirical evaluation:** We perform rigorous benchmarking incorporating state-of-the-art graph condensation algorithms on 7 real-world datasets containing up to hundreds of millions of edges. Our analysis establishes that BONSAI *(1)* achieves higher prediction accuracy, *(2)* produces at least 7-times faster condensation times despite being CPU-bound in contrast to GPU-bound condensation of baselines, and *(3)* exhibits superior robustness to GNN architectures and datasets.

---

[1]Some algorithms sparsify the fully-connected graph based on edge weights. But this sparsification process requires training on the fully connected graph itself to identify the pruning threshold.

[2]Inspired by the art of Bonsai, which transforms large trees into miniature forms while preserving their essence, our graph condensation algorithm gracefully prunes redundant computation trees, creating a condensed graph that is significantly smaller yet maintains comparable performance.

## 2 PROBLEM FORMULATION AND PRELIMINARIES

**Definition 1** (Graph). *$\mathcal{G} = (\mathcal{V}, \mathcal{E}, \boldsymbol{X})$ denotes a graph over a finite, non-empty node set $\mathcal{V}$ and edge set $\mathcal{E} = \{(u, v) \mid u, v \in \mathcal{V}\}$. $\boldsymbol{X} \in \mathbb{R}^{|\mathcal{V}| \times |F|}$ denotes node attributes encoded using $F$-dimensional feature vectors. We denote the attributes of node $v$ as $\mathbf{x}_v$.*

Two graphs are identical if they are *isomorphic* to each other.

**Definition 2** (Graph Isomorphism). *Graph $\mathcal{G}_1$ is isomorphic to graph $\mathcal{G}_2$ if there exists a bijective mapping between their node sets that preserves both edge connectivity and node features. Specifically, $\mathcal{G}_1$ is isomorphic to $\mathcal{G}_2 \iff \exists f : \mathcal{V}_1 \to \mathcal{V}_2$ such that: (1) $f$ is a bijection, (2) $\mathbf{x}_v = \mathbf{x}_{f(v)}$, where $v \in \mathcal{V}_1, f(v) \in \mathcal{V}_2$ and (3) $(u, v) \in \mathcal{E}_1$ if and only if $(f(u), f(v)) \in \mathcal{E}_2$.*

The problem of graph condensation for node classification is defined as follows.

**Problem 1** (Graph Condensation). *Given train and validation graphs, $\mathcal{G}_{tr}$ and $\mathcal{G}_{val}$, respectively, and a memory budget $b$ in bytes, synthesize a graph $\mathcal{G}_s$ from $\mathcal{G}_{tr}$ within the budget, while minimizing the error gap between $\mathcal{G}_s$ and $\mathcal{G}_{tr}$ on the validation set, i.e., minimize $\{|\epsilon_{\mathcal{G}_s} - \epsilon_{\mathcal{G}_{tr}}|\}$. $\epsilon_{\mathcal{G}}$ represents the node classification error on the validation set when trained on graph $\mathcal{G}$.*

### 2.1 COMPUTATION STRUCTURE OF GNNS

GNNs operate through an iterative process of information exchange between nodes. Let $\mathbf{x}_v \in \mathbb{R}^{|F|}$ represent the initial feature vector of node $v \in \mathcal{V}$. The propagation mechanism proceeds as follows:

**Initialization:** Set $\mathbf{h}_v^0 = \mathbf{x}_v, \forall v \in \mathcal{V}$.

**Message creation:** In layer $\ell$, *collect* and *aggregate* messages for each neighbor.

$$\mathbf{m}_v^\ell(u) = \text{MESSAGE}^\ell(\mathbf{h}_u^{\ell-1}, \mathbf{h}_v^{\ell-1}), \quad \forall u \in \mathcal{N}_v = \{u \mid (u, v) \in \mathcal{E}\}$$

$$\overline{\mathbf{m}}_v^\ell = \text{AGGREGATE}^\ell(\{\!\!\{\mathbf{m}_v^\ell(u) : u \in \mathcal{N}_v\}\!\!\})$$

**Update embedding:** $\mathbf{h}_v^\ell = \text{UPDATE}^\ell(\mathbf{h}_v^{\ell-1}, \overline{\mathbf{m}}_v^\ell)$

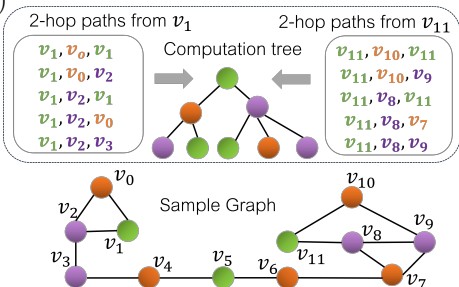

Here, $\text{MESSAGE}^\ell$, $\text{AGGREGATE}^\ell$, and $\text{UPDATE}^\ell$ may be predefined operations (e.g., mean pooling) or learnable neural networks. $\{\!\!\{\cdot\}\!\!\}$ denotes a multiset. This process repeats for $L$ layers, yielding the final node representations $\mathbf{h}_v^L$.

A *computation tree* describes how information propagates through the graph during the neural network's operation (Xu et al., 2019).

**Definition 3** (Computation Tree). *Given a graph $\mathcal{G} = (\mathcal{V}, \mathcal{E}, \boldsymbol{X})$, a node $v \in \mathcal{V}$ and the number of layers $L$ in the GNN, the computation tree $T_v^L$ rooted at $v$ is constructed as follows:*

- *Enumerate all paths of length $L$ (including those with repeated vertices) starting from $v$.*
- *Merge these paths into a tree structure where:*
  1. *The root is always $v$, and*
  2. *Two nodes from different paths are merged if: **(i)** They are at the same depth in their respective paths, and **(ii)** All their ancestors in the paths have already been merged.*

Figure 1: The figure depicts the construction of computation trees for nodes $v_1$ and $v_{11}$ in the sample graph, at depth $L = 2$. Node colors indicate their labels. Despite being distant from each other in the graph and embedded in non-isomorphic $L$-hop neighborhoods, $v_1$ and $v_{11}$ have isomorphic computation trees.

Fig. 1 illustrates the idea of a computation tree with an example.

**Properties of computation trees:** We now highlight two key properties of computation trees, formally established in Xu et al. (2019), that form the core of our algorithm design. These properties hold regardless of the underlying message-passing GNN (GCN, GAT, GRAPHSAGE, GIN, etc.).

**Property 1** (Sufficiency). *In an $L$-layered GNN, the computation tree $\mathcal{T}_v^L$ is sufficient to compute node embedding $\mathbf{h}_v^L$, $\forall v \in \mathcal{V}$. Hence, given a graph $\mathcal{G} = (\mathcal{V}, \mathcal{E}, \boldsymbol{X})$, we may treat it as a (multi)set of computation trees $\mathbb{T} = \{\mathcal{T}_v^L \mid \forall v \in \mathcal{V}\}$. Clearly, $|\mathbb{T}| = |\mathcal{V}|$.*

**Property 2** (Equivalence). *If $\mathcal{T}_v^L$ is isomorphic to $\mathcal{T}_u^L$, then $\mathbf{h}_v^L = \mathbf{h}_u^L$ since the expressive power of a message-passing GNN is upper bounded by the Weisfeiler-Lehman test (1-WL) (Xu et al., 2019).*

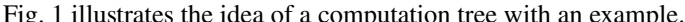

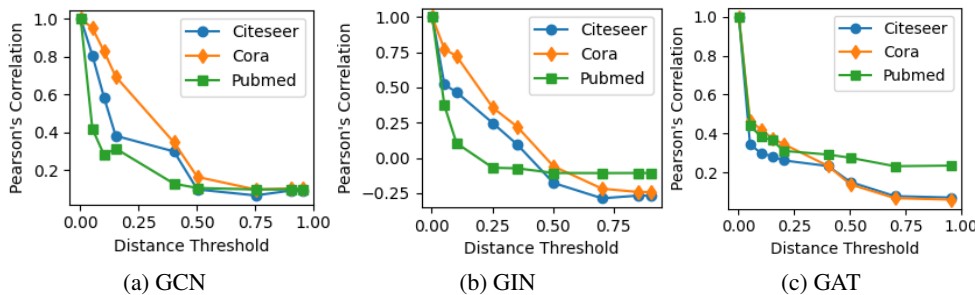

Figure 2: Pearson correlations between the $L_2$ distances of node pairs in the GNN embedding space and the unsupervised embeddings derived from the WL-kernel, computed for pairs with a specific distance threshold in the WL-space ($x$-axis).

Motivated with the above observations, we explore a relaxation of the Equivalence property: *do nodes rooted at topologically similar computation trees generate similar node embeddings?* To explore this hypothesis, we need a method to quantify the distance between computation trees. Given that the Equivalence property is derived from the 1-WL test, the *Weisfeiler-Lehman kernel* (Togninalli et al., 2019) presents itself as the natural choice for this distance metric.

**Definition 4** (Weisfeiler-Lehman (WL) Kernel (Togninalli et al., 2019)). *Given graph* $\mathcal{G} = (\mathcal{V}, \mathcal{E}, \boldsymbol{X})$, *WL-kernel constructs unsupervised embeddings over each node in the graph via a message-passing aggregation. Like in GNNs, the initial embedding* $\mathbf{a}_v^0$ *in layer* $0$ *is initialized to* $\mathbf{x}_v$. *Subsequently, the embeddings in any layer* $\ell$ *is defined as:*

$$\mathbf{a}^\ell(v) = \frac{1}{2}\left(\mathbf{a}_v^{\ell-1} + \frac{1}{\deg(v)} \sum_{u \in \mathcal{N}(v)} w((v,u)) \cdot \mathbf{a}_u^{\ell-1}\right) \tag{1}$$

Here, $\mathcal{N}(v)$ denotes the neighbors of $v$, $\deg(v) = |\mathcal{N}(v)|$ and $w((v,u)) = 1$ for unweighted graphs.

By emulating the same message-passing framework employed by GNNs, the unsupervised embedding $\mathbf{a}_v^L$ jointly encapsulates the topology and node attributes within the computation tree $\mathcal{T}_v^L$. Hence, the distance between two computation trees $\mathcal{T}_v^L$ and $\mathcal{T}_u^L$ is defined to be:

$$d\left(\mathcal{T}_v^L, \mathcal{T}_v^L\right) = \left\|\mathbf{a}_v^L - \mathbf{a}_u^L\right\|_2 \tag{2}$$

While we use the $L_2$ norm, one may use other distance functions over vectors as well. The following corollary follows from the Eq. 2.

**Corollary 1.** *If* $\mathcal{T}_v^L$ *is isomorphic to* $\mathcal{T}_u^L$, *then* $\mathbf{a}_v^L = \mathbf{a}_u^L$.

**Hypothesis 1** (Relaxed Equivalence). *If* $\mathcal{T}_v^L \sim \mathcal{T}_u^L$, *then* $\mathbf{a}_v^L \sim \mathbf{a}_u^L$, *and, therefore* $\mathbf{h}_v^L \sim \mathbf{h}_u^L$ *irrespective of the specific message-passing architecture employed.*

To test our hypothesis, in Fig. 2, we examine all node pairs within a specified distance threshold based on the Weisfeiler-Lehman (WL) kernel. We then analyze the correlation between their unsupervised WL-distance and the $L_2$ distance between their GNN embeddings. The results reveal a compelling trend: as we decrease the distance threshold, we observe a strengthening correlation between WL-distance and GNN embedding distance. This pattern supports our hypothesis, indicating that computation trees that are proximate in the WL-space also exhibit proximity in the GNN embedding space.

## 3  BONSAI: PROPOSED METHODOLOGY

Fig. 3 presents the pipeline of BONSAI, which is founded on the following logical progression:

1. Similar computation trees produce similar GNN embeddings (Hypotheses 1).
2. Similar GNN embeddings generate comparable outputs, resulting in similar impacts on the loss function and, consequently, on the gradients Table 2 presents empirical data supporting this relationship, showing statistically significant correlations ($p < 0.05$) between WL-embedding similarities and training gradients.

Table 2: Correlation between WL-embedding similarities and training gradients.

| Dataset | Correlation | $p$-value |
|---|---|---|
| Cora | 0.74 | $\approx 0$ |
| Citeseer | 0.83 | $\approx 0$ |
| Pubmed | 0.38 | 0.02 |
| Reddit | 0.42 | $\approx 0$ |

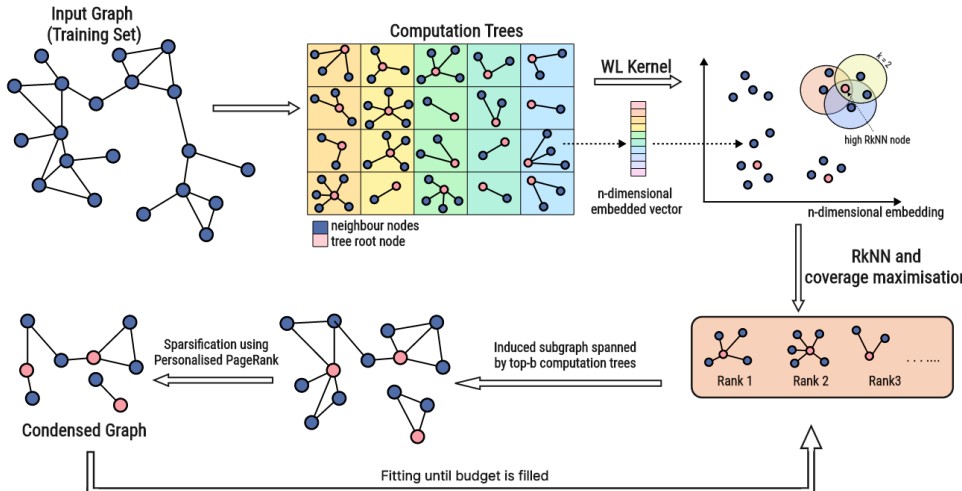

Figure 3: Pipeline of the proposed algorithm for graph condensation using a toy example is displayed. We use $L = 1$ and $k = 2$ for this example.

Building on this reasoning, BONSAI aims to identify a set of $b$ *exemplar* computation trees that optimally represent the full training set[1]. These exemplars are selected based on two critical criteria:

- **Representativeness:** Each exemplar should be similar to a large number of other computation trees from the input training set, ensuring it captures common structural patterns. We quantify the representative power of a computation tree using the idea of *reverse $k$ nearest neighbors* (§ 3.1). Specifically, if tree $T_1$ is among the $k$-NN of tree $T_2$ for a small $k$, this indicates these two trees are similar in WL-embedding. Hence, we seek to include those trees in the condensed set that reside in the $k$-NN of lots of other trees. Consequently, if these trees are selected, their GNN embeddings are also likely similar to their $k$-NN neighbors from the WL space. As a result, they can effectively approximate the GNN embeddings of the filtered-out nodes. Since similar GNN embeddings lead to similar gradients (Table 2), we minimize the information lost from nodes that are filtered out.
- **Diversity:** The set of exemplars should be diverse, maximizing the coverage of different structural patterns present in the original graph (§ 3.2). To achieve this objective, we develop a greedy tree selection algorithm (Alg. 1), which begins with the reverse $k$-NN set and iteratively selects the computation tree that appears in the $k$-NN sets of the maximum number of currently uncovered trees. This approach ensures two key properties:
  1. It systematically captures trees that are centrally located within the graph's embedding space.
  2. It progressively selects trees that provide maximum coverage of the remaining, yet-unselected computation trees.

By prioritizing trees with the highest marginal impact on uncovered trees, our method naturally creates a diverse subset that comprehensively represents the original graph's structural and computational characteristics. The induced subgraph spanned by these exemplar trees forms the initial condensed set. This initial version undergoes further refinement through an iterative process of *edge sparsification* and *enrichment* to reduce $O(|\mathcal{E}|)$ complexity of GNNs.

### 3.1 Quantifying Representativeness through Reverse $k$-NN

Utilizing Sufficiency (Property 1), we decompose the input graph $\mathcal{G} = (\mathcal{V}, \mathcal{E}, \boldsymbol{X})$ into a set of $|\mathcal{V}|$ computation trees $\mathbb{T}$ and then embed them into a feature space using WL-kernel (See Fig. 3). We now identify representative exemplars by analyzing this space.

The $k$ nearest neighbors of a computation tree $\mathcal{T}_v^L \in \mathbb{T}$, denoted as $k$-NN$(\mathcal{T}_v^L)$, are its $k$ nearest computations trees from $\mathbb{T}$ in the WL-kernel space (Def. 4).

**Definition 5** (Reverse $k$-NN and Representative Power). *The reverse $k$-NN of $\mathcal{T}_v^L$ denotes the set of trees that contains $\mathcal{T}_v^L$ in their $k$-NNs. Formally, Rev-$k$-NN$(\mathcal{T}_v^L) = \left\{ \mathcal{T}_u^L \mid \mathcal{T}_v^L \in k\text{-NN}\left(\mathcal{T}_u^L\right) \right\}$. The representative power of $\mathcal{T}_v^L$ is:*

$$\Pi\left(\mathcal{T}_v^L\right) = \frac{\left|Rev\text{-}k\text{-}NN(\mathcal{T}_v^L)\right|}{|\mathbb{T}|} \tag{3}$$

---

[1]We overload the notation for the budget by denoting the number of exemplar trees that fit within the input budget constraint (Recall Problem. 1) as $b$.

If $\mathcal{T}_v^L \in k\text{-NN}(\mathcal{T}_u^L)$ for a small $k$, it suggests that $\mathbf{h}_v^L \sim \mathbf{h}_u^L$ (Hypothesis 1). Consequently, a high $\Pi\left(\mathcal{T}_v^L\right)$ indicates that $\mathcal{T}_v^L$ frequently appears in the $k$-NN lists of many other trees and is therefore positioned in a dense region capturing shared characteristics across many trees. Hence, it serves as a strong candidate of being an exemplar.

### 3.1.1 Sampling for Scalable Computation of Reverse $k$-NN

Computing $k$-NN for each tree consumes $\mathcal{O}(n \log k) \approx \mathcal{O}(n)$ time (since $k \ll n$), where $n = |\mathcal{V}|$. Since $|\mathbb{T}| = n$, computing $k$-NN for all trees consumes $\mathcal{O}(n^2)$ time. Hence, computing reverse $k$-NN for all trees consumes $\mathcal{O}(n^2)$ time as well, which may be prohibitively expensive for graphs containing millions of nodes. To address this challenge, we employ a *sampling* technique that offers a provable approximation guarantee on accuracy. This is achieved as follows. Let us sample $z \ll n$ trees uniformly at random from $\mathbb{T}$, which we denote as $\mathbb{S}$. Now, we compute the $k$-NN of only trees in $\mathbb{S}$, which incurs $\mathcal{O}(zn)$ computation cost. We next approximate reverse $k$-NN of all trees in $\mathbb{T}$ based only on $\mathbb{S}$. Specifically, $\widetilde{\text{Rev-}k\text{-NN}}\left(\mathcal{T}_v^L\right) = \left\{\mathcal{T}_u^L \in \mathbb{S} \mid \mathcal{T}_v^L \in k\text{-NN}\left(\mathcal{T}_u^L\right)\right\}$. The approximated representative power is therefore:

$$\widetilde{\Pi}\left(\mathcal{T}_v^L\right) = \frac{\left|\widetilde{\text{Rev-}k\text{-NN}}(\mathcal{T}_v^L)\right|}{|\mathbb{S}|} \tag{4}$$

The sample size $z$ balances the trade-off between accuracy and computational efficiency. By applying *Chernoff bounds*, we demonstrate in Lemma 1 that $z$ can be precisely determined to maintain the error within a specified error threshold $\theta$, with a confidence level of $1 - \delta$.

**Lemma 1.** *Given a desirable error upper bound $\theta$ and a confidence interval of $1 - \delta$, if we sample at least $z = \frac{\ln\left(\frac{2}{\delta}\right)(2+\theta)}{\theta^2}$ trees in $\mathbb{S}$, then for any $\mathcal{T}_v^L \in \mathbb{T}$:*

$$P\left(\left|\widetilde{\Pi}(\mathcal{T}_v^L) - \Pi\left(\mathcal{T}_v^L\right)\right| \leq \theta\right) \geq 1 - \delta \tag{5}$$

PROOF. For the formal proof, see App. A.1. Lemma 1 induces the following positive implications.
- The number of samples needed is independent of the size of $\mathbb{T}$.
- Because $z$ grows logarithmically with $ln(\frac{2}{\delta})$, even a small sample size provides high confidence. Hence, the computation cost of Rev-$k$-NN reduces to $\mathcal{O}(n)$ since $z \ll n$.

### 3.2 Coverage Maximization

We aim to find the set of *exemplar* computation trees with the maximum representative power.

**Definition 6** (The exemplars). *Let the representative power of a set of trees $\mathbb{A}$ be denoted by:*

$$\Pi\left(\mathbb{A}\right) = \left|\bigcup_{\forall \mathcal{T}_v^L \in \mathbb{A}} Rev\text{-}k\text{-}NN\left(\mathcal{T}_v^L\right)\right| \Big/ |\mathbb{T}| \tag{6}$$

*Then, given the set of computation trees $\mathbb{T}$ and the budget $b$, we seek to identify the subset of computations trees, termed exemplars, $\mathbb{A}^*$, by maximizing the following objective:*

$$\mathbb{A}^* = \max_{\forall \mathbb{A} \subseteq \mathbb{T}, |\mathbb{A}| = b} \Pi(\mathbb{A}) \tag{7}$$

For brevity of discourse, we proceed assuming the true reverse $k$-NN set.

**Theorem 1.** *Maximizing the representative power of exemplars (Eq. 7) is NP-hard.*

PROOF. App. A.2 presents the formal proof by reducing to the *Set Cover problem* (Cormen et al., 2009). □

Fortunately, Eq. 6 is *monotone* and *submodular*, which allows the greedy hill-climbing algorithm (Alg. 1) to approximate Eq. 7 within a provable error bound.

**Theorem 2.** *The exemplars, $\mathbb{A}_{greedy}$, selected by Alg. 1 provides an $1 - 1/e$ approximation, i.e., $\Pi(\mathbb{A}_{greedy}) \geq \left(1 - \frac{1}{e}\right)\Pi(\mathbb{A}^*)$.*

---

**Algorithm 1** The greedy approach

**Require:** Graph $\mathcal{G}$, budget $b$, $Rev-\mathrm{k}-NN\left(\mathcal{T}_v^L\right)$
**Ensure:** solution set $\mathbb{A}$, $|\mathbb{A}| = b$
1: $\mathbb{A} \leftarrow \emptyset$
2: **while** $size(\mathbb{A}) \leq b$ (within budget) **do**
3: $\quad \mathcal{T}_{v*}^L \leftarrow \arg\max_{\mathcal{T}_v^L \in \mathbb{T} \setminus \mathbb{A}} \{\Pi(\mathbb{A} \cup \{\mathcal{T}_v^L\}) - \Pi(\mathbb{A})\}$
4: $\quad \mathbb{A} \leftarrow \mathbb{A} \cup \{\mathcal{T}_{v*}^L\}$
5: **Return** $\mathbb{A}$

---

PROOF. App. A.3 presents the proofs of monotonicity and submodularity of $\Pi(\mathbb{A})$. For monotonic and submodular functions, greedy selection provides $1 - 1/e$ approximation (Feige, 1998).                □.

Alg. 1 begins with the reverse $k$-NN set of each computation tree as input. It then iteratively selects trees based on their *marginal cardinality* - specifically, choosing the tree that appears in the $k$-NN sets of the largest number of yet-uncovered trees (lines 3-4). A tree is considered uncovered if none of its $k$-nearest neighbors have been selected for the condensed set. This focus on marginal contribution naturally promotes diversity. Consider two similar trees, $T_1$ and $T_2$, both with high reverse $k$-NN cardinality. Due to the transitivity of distance functions, these trees likely share many of the same neighbors in their reverse $k$-NN sets. Consequently, if $T_1$ is selected, $T_2$'s marginal cardinality significantly decreases despite its high initial reverse $k$-NN cardinality, preventing redundant selection of similar trees.

We further enhance the efficiency of Alg. 1 by exploiting the property of monotonically decreasing *marginal gains* in submodular optimization (Leskovec et al., 2007).

**Initial condensed graph:** The initial condensed graph $\mathcal{G}_s = (\mathcal{V}_s, \mathcal{E}_s, \boldsymbol{X}_s)$ is formed by extracting the induced subgraph spanned by the exemplar computation trees, i.e., $\mathcal{V}_s = \left\{ u \mid \exists \mathcal{T}_v^L \in \mathbb{A}_{greedy}, \ u \in \mathcal{N}_L(v) \right\}, \mathcal{E}_s = \{(u, v) \mid u \in \mathcal{V}_s, v \in \mathcal{V}_s\}$ and $\boldsymbol{X}_s = \{\mathbf{x}_v) \mid v \in \mathcal{V}_s\}$. Set $\mathcal{N}_L(v)$ contains nodes within $L$ hops from $v$.

**Sparsification and Enrichment:** In the final step, we seek to sparsify the graph induced by the exemplar trees. Furthermore, in the additional space created due to sparsfication, we include more exemplar trees, resulting in further magnification of the representative power. Further details of our implementation is provided in App. A.4.

### 3.3 PROPERTIES OF BONSAI

**Complexity analysis:** The computation complexity of BONSAI is $\mathcal{O}(|\mathcal{V}| + |\mathcal{E}|)$ (details in App. A.5).

**CPU-bound and Parallelizable:** BONSAI does not involve learning any parameters through gradient descent or its variants. Hence, the entire process is CPU-bound. Furthermore, all steps are embarrassingly parallelizable leading condensation of datasets with hundreds of millions of edges within minutes.

**Independence from model-architecture and hyper-parameters:** Unlike majority of existing condensation algorithms that require knowledge of the GNN architecture and all training hyper-parameters, BONSAI only requires approximate knowledge of the number of layers $L$.

## 4 EXPERIMENTS

In this section, we benchmark BONSAI and establish:

- **Superior accuracy:** BONSAI consistently outperforms existing baselines in terms of accuracy across various compression factors, datasets, and GNN architectures.
- **Enhanced computation and energy efficiency:** On average, BONSAI is at least 7 times faster and 17 times more energy efficient than the state-of-the-art baselines.
- **Increased robustness:** Unlike existing methods that require tuning condensation-specific hyper-parameters for each combination of GNN architecture, dataset, and compression ratio, BONSAI achieves superior performance using a single set of parameters across all scenarios.

Our implementation is available at https://github.com/idea-iitd/Bonsai.

### 4.1 EXPERIMENTAL SETUP

The specifics of our experimental setup, including hardware and software environment, and hyper-parameters are detailed in App. B. For the baseline algorithms, we use the code shared by their respective authors. We conduct each experiment 5 times and report the means and standard deviations.

Table 3: Datasets.

| Dataset | # Nodes | # Edges | # Classes | # Features |
|---|---|---|---|---|
| Cora (Kipf & Welling, 2017) | 2,708 | 10,556 | 7 | 1,433 |
| Citeseer (Kipf & Welling, 2017) | 3,327 | 9,104 | 6 | 3,703 |
| Pubmed (Kipf & Welling, 2017) | 19,717 | 88,648 | 3 | 500 |
| Flickr (Zeng et al., 2020) | 89,250 | 899,756 | 7 | 500 |
| Ogbn-arxiv (Hu et al., 2021) | 169,343 | 2,315,598 | 40 | 128 |
| Reddit (Hamilton et al., 2017) | 232,965 | 23,213,838 | 41 | 602 |
| MAG240M (Hu et al., 2021) | 1,398,159 | 26,434,726 | 153 | 768 |

**Datasets:** Table 3 lists the benchmark datasets used.

Table 4: Accuracy achieved by the various baselines on benchmark datasets across various compression ratios over byte consumption (denoted as $\mathbf{S_r}(\%)$), on the GCN architecture. The best and the second-best accuracies in each row are highlighted by dark and lighter shades of **Green**, respectively. OOT indicates the scenario where the algorithm failed to condense within 5 hours.

| Dataset | $S_r(\%)$ | Random | Herding | GCond | GDEM | GCSR | EXGC | GCSNTK | GEOM | Bonsai | Full |
|---|---|---|---|---|---|---|---|---|---|---|---|
| Cora | 0.5 | 39.90±1.46 | 45.61±0.01 | 77.30±0.30 | 57.49±6.87 | 74.83±0.75 | 34.87±0.83 | 77.85±0.76 | 33.76±0.96 | 83.95±0.39 | 88.56±0.18 |
| | 1 | 27.75±2.05 | 52.07±0.00 | 77.30±0.31 | 69.32±4.71 | 77.56±0.74 | 35.24±0.60 | 67.52±0.77 | 33.02±0.92 | 85.76±0.24 | |
| | 3 | 52.26±1.69 | 67.60±0.00 | 81.73±0.48 | 81.70±3.10 | 77.20±0.48 | 35.42±0.77 | 75.64±0.82 | 54.80±1.89 | 86.38±0.22 | |
| CiteSeer | 0.5 | 33.90±2.16 | 22.82±0.00 | 74.17±0.68 | 70.05±2.40 | 67.03±0.61 | 23.42±0.98 | 69.82±0.68 | 24.92±0.89 | 77.00±0.15 | 78.53±0.15 |
| | 1 | 44.90±2.32 | 49.10±0.02 | 77.62±0.71 | 72.48±2.13 | 74.77±0.78 | 23.67±1.00 | 69.22±0.71 | 28.03±0.81 | 77.03±0.33 | |
| | 3 | 44.50±1.27 | 67.69±0.01 | 77.02±0.22 | 76.20±0.55 | 77.27±0.28 | 25.07±0.92 | 64.26±0.53 | 33.48±0.83 | 75.89±0.26 | |
| Pubmed | 0.5 | 62.58±0.25 | 78.29±0.00 | 80.63±1.20 | 80.72±0.92 | 79.43±0.25 | 45.77±0.73 | 53.04±1.99 | OOT | 87.27±0.03 | 87.22±0.00 |
| | 1 | 79.19±0.09 | 78.59±0.00 | 79.92±0.00 | 80.80±1.07 | 79.11±0.15 | 46.24±0.43 | 62.81±1.32 | OOT | 87.08±0.04 | |
| | 3 | 82.50±0.09 | 78.09±0.00 | 77.00±0.15 | 81.07±0.90 | 79.94±0.16 | 47.62±0.67 | 67.72±2.01 | OOT | 87.64±0.09 | |
| Flickr | 0.5 | 44.78±0.00 | 47.98±0.01 | 44.06±1.05 | 46.25±1.02 | 46.41±0.00 | 45.47±0.85 | 31.49±0.75 | OOT | 48.73±0.27 | 50.93±0.17 |
| | 1 | 44.21±0.03 | 46.72±0.01 | 39.88±5.60 | 46.99±1.38 | OOT (5 hrs) | 45.97±0.82 | 42.50±0.99 | OOT | 49.05±0.17 | |
| | 3 | 46.56±0.01 | 46.54±0.01 | 46.04±1.88 | 47.35±0.98 | OOT | 48.44±0.65 | 36.58±0.82 | OOT | 49.66±0.27 | |
| Ogbn-arxiv | 0.5 | 42.01±0.41 | 53.37±0.00 | 52.63±0.63 | 54.73±0.66 | OOT | 60.66±1.66 | 61.55±1.17 | OOT | 58.49±0.17 | 68.97±0.10 |
| | 1 | 49.27±0.64 | 54.91±0.00 | 53.49±0.63 | 51.45±1.14 | OOT | 61.73±1.43 | 62.31±0.88 | OOT | 58.35±0.09 | |
| | 3 | 51.11±0.19 | 57.28±0.00 | 53.01±0.64 | 53.37±1.04 | OOT | 62.96±1.33 | 56.38±0.79 | OOT | 64.31±0.06 | |
| Reddit | 0.5 | 36.00±4.09 | 81.72±0.63 | 38.94±0.79 | 90.51±0.55 | OOT | 78.47±0.52 | 37.15±1.51 | OOT | 80.33±0.46 | 92.14±0.04 |
| | 1 | 38.55±2.00 | 83.48±0.83 | 43.98±0.35 | 90.63±0.84 | OOT | 81.69±1.12 | 38.87±2.00 | OOT | 85.65±0.08 | |
| | 3 | 44.97±2.97 | 88.51±0.13 | 48.78±0.83 | 85.75±0.80 | OOT | OOM | 47.48±1.98 | OOT | 88.90±0.07 | |
| MAG240M | 0.5 | 34.43±0.11 | 36.18±0.10 | OOT | OOM | OOT | OOM | OOT | OOT | 52.33±0.05 | 69.95±3.52 |
| | 1 | 37.93±0.08 | 37.29±0.06 | OOT | OOM | OOT | OOM | OOT | OOT | 52.58±0.33 | |
| | 3 | 42.96±0.43 | 37.98±0.09 | OOT | OOM | OOT | OOM | OOT | OOT | 53.39±0.07 | |

**Baselines:** Table 1 lists all the graph condensation algorithms for node classification. We omit SFGC (Zheng et al., 2023) and SGDD (Yang et al., 2023) as baselines since both GDEM and GCSR have been shown to outperform them. Among non-neural, baselines, we compare with selecting the induced subgraph spanned by a random selection of nodes, and Herding (Welling, 2009).

**Metrics:** Prediction quality is measured through Accuracy on the test set, i.e., the percentage of correct predictions. Compression ratio is quantified as $S_r = \frac{\text{size of condensed dataset in bytes}}{\text{size of full data set in bytes}}$.

## 4.2 PREDICTION ACCURACY

Table 4 presents the accuracies obtained by BONSAI and the baselines on GCN. BONSAI achieves the best accuracy in 12 out of 18 scenarios, with substantial improvements ($\geq 5\%$) over the second-best performer in several cases. This demonstrates that the unsupervised gradient-agnostic approach adopted by BONSAI does not come at the cost of accuracy. GDEM emerges as the second best perfomer. GCSR fails to complete in the two largest datasets of Ogbn-arxiv and Reddit, since it trains on the entire dataset 100 times to condense, which exceeds 5 hours. GEOM is even slower since it trains on full train set 200 times. All baselines except Random and Herding failed to finish within 5 hours on MAG240M.

## 4.3 CROSS-ARCHITECTURE GENERALIZATION

GDEM and BONSAI are both model-agnostic, meaning their condensed datasets are independent of the downstream GNN architecture used. In contrast, GCOND and GCSR produce architecture-specific condensation datasets, requiring separate training for each architecture. This distinction raises two important questions: First, *how well do the condensed datasets generated by GDEM and BONSAI generalize across different architectures?* Second, if we apply the condensation datasets produced by GCOND and GCSR for GCN to other architectures, how significantly does it impact performance? We explore these questions in Table 5.

Consistent with the earlier trend, BONSAI continues to outperform all baselines. We further observe that the performance gap between BONSAI and the baselines is wider in GAT and GIN compared to GCN (Table 4). For GCOND and GCSR, this wider gap is not surprising since they are trained on the gradients of GCN, which are expected to differ from those of GIN and GAT. GIN uses a Sum-Pool aggregation, while GAT employs attention to effectively dampen unimportant edges, with a potentially different eigen spectrum - the property that GDEM attempts to preserve. The approach of BONSAI, on the other hand, aligns more directly with the computational structure of GNNs. It focuses on analyzing the input space of computation trees and aims to represent as much of this space as possible within the given budget.

Table 5: Accuracies achieved by the various baselines on GAT and GIN.

| Dataset | % | GNN | Random | Herding | GCond | GDEM | GCSR | BONSAI | Full |
|---|---|---|---|---|---|---|---|---|---|
| Cora | 0.5 | GAT | $41.44_{\pm1.73}$ | $33.80_{\pm0.07}$ | $13.21_{\pm1.99}$ | $63.91_{\pm5.91}$ | $15.09_{\pm6.19}$ | $75.42_{\pm1.61}$ | |
| | 1 | GAT | $42.73_{\pm1.03}$ | $46.09_{\pm0.86}$ | $35.24_{\pm0.00}$ | $73.49_{\pm2.64}$ | $37.60_{\pm1.34}$ | $78.67_{\pm0.89}$ | $85.70_{\pm0.09}$ |
| | 3 | GAT | $60.22_{\pm0.67}$ | $56.75_{\pm0.45}$ | $35.24_{\pm0.00}$ | $75.28_{\pm4.86}$ | $36.72_{\pm0.81}$ | $80.66_{\pm0.80}$ | |
| | 0.5 | GIN | $49.04_{\pm0.50}$ | $34.39_{\pm1.03}$ | $14.13_{\pm6.80}$ | $63.65_{\pm7.11}$ | $76.05_{\pm0.44}$ | $85.42_{\pm0.74}$ | |
| | 1 | GIN | $50.48_{\pm0.85}$ | $33.80_{\pm2.42}$ | $33.91_{\pm1.23}$ | $75.92_{\pm4.24}$ | $60.70_{\pm4.44}$ | $84.80_{\pm0.41}$ | $86.62_{\pm0.28}$ |
| | 3 | GIN | $59.52_{\pm0.88}$ | $36.35_{\pm0.59}$ | $31.70_{\pm4.97}$ | $59.59_{\pm7.95}$ | $51.62_{\pm5.00}$ | $85.42_{\pm0.53}$ | |
| CiteSeer | 0.5 | GAT | $42.76_{\pm0.35}$ | $36.04_{\pm0.46}$ | $21.47_{\pm0.00}$ | $69.86_{\pm2.28}$ | $21.92_{\pm0.76}$ | $68.56_{\pm0.57}$ | |
| | 1 | GAT | $46.19_{\pm1.38}$ | $52.07_{\pm0.11}$ | $21.47_{\pm0.00}$ | $23.87_{\pm3.05}$ | $21.50_{\pm0.06}$ | $69.43_{\pm0.82}$ | $77.48_{\pm0.75}$ |
| | 3 | GAT | $61.65_{\pm0.51}$ | $65.17_{\pm0.00}$ | $21.26_{\pm0.22}$ | $22.90_{\pm1.20}$ | $21.50_{\pm0.06}$ | $69.94_{\pm1.15}$ | |
| | 0.5 | GIN | $44.86_{\pm0.43}$ | $22.97_{\pm0.30}$ | $21.47_{\pm0.00}$ | $67.69_{\pm3.28}$ | $50.66_{\pm1.17}$ | $71.80_{\pm0.26}$ | |
| | 1 | GIN | $47.90_{\pm0.65}$ | $39.67_{\pm0.82}$ | $19.49_{\pm1.09}$ | $67.64_{\pm4.45}$ | $64.74_{\pm1.88}$ | $72.16_{\pm0.60}$ | $75.45_{\pm0.23}$ |
| | 3 | GIN | $61.83_{\pm0.68}$ | $60.48_{\pm0.26}$ | $18.65_{\pm2.56}$ | $48.65_{\pm8.17}$ | $59.95_{\pm9.07}$ | $70.51_{\pm0.54}$ | |
| PubMed | 0.5 | GAT | $77.73_{\pm0.12}$ | $75.44_{\pm0.02}$ | $37.49_{\pm4.01}$ | $80.06_{\pm1.16}$ | $38.29_{\pm8.13}$ | $85.66_{\pm0.38}$ | |
| | 1 | GAT | $78.85_{\pm0.09}$ | $76.64_{\pm0.02}$ | $41.55_{\pm3.18}$ | $80.75_{\pm0.47}$ | $40.47_{\pm0.00}$ | $85.88_{\pm0.28}$ | $86.33_{\pm0.08}$ |
| | 3 | GAT | $82.84_{\pm0.11}$ | $78.48_{\pm0.03}$ | $37.77_{\pm3.61}$ | $65.08_{\pm9.53}$ | $40.27_{\pm0.20}$ | $85.62_{\pm0.36}$ | |
| | 0.5 | GIN | $77.45_{\pm0.14}$ | $48.48_{\pm1.33}$ | $30.91_{\pm4.57}$ | $78.78_{\pm0.91}$ | $36.88_{\pm12.06}$ | $84.32_{\pm0.33}$ | |
| | 1 | GIN | $78.43_{\pm0.22}$ | $62.22_{\pm0.13}$ | $32.84_{\pm6.27}$ | $78.72_{\pm0.95}$ | $33.75_{\pm5.58}$ | $85.57_{\pm0.26}$ | $84.66_{\pm0.05}$ |
| | 3 | GIN | $80.56_{\pm0.17}$ | $45.40_{\pm0.46}$ | $36.11_{\pm3.47}$ | $81.08_{\pm0.99}$ | $32.01_{\pm6.77}$ | $85.66_{\pm0.23}$ | |
| Flickr | 0.5 | GAT | $43.64_{\pm0.99}$ | $36.50_{\pm13.22}$ | $40.24_{\pm3.20}$ | $25.43_{\pm10.37}$ | $28.03_{\pm6.60}$ | $48.22_{\pm3.60}$ | |
| | 1 | GAT | $43.56_{\pm1.06}$ | $36.34_{\pm1.14}$ | $40.85_{\pm1.08}$ | $18.44_{\pm9.42}$ | OOT | $45.62_{\pm1.85}$ | $51.42_{\pm0.07}$ |
| | 3 | GAT | $45.71_{\pm1.87}$ | $42.70_{\pm1.17}$ | $41.51_{\pm9.81}$ | $25.83_{\pm11.39}$ | OOT | $47.80_{\pm2.06}$ | |
| | 0.5 | GIN | $42.67_{\pm0.83}$ | $39.98_{\pm7.21}$ | $13.65_{\pm7.54}$ | $14.10_{\pm5.68}$ | $05.92_{\pm1.01}$ | $44.97_{\pm2.23}$ | |
| | 1 | GIN | $42.90_{\pm0.76}$ | $41.87_{\pm4.52}$ | $16.65_{\pm6.55}$ | $19.44_{\pm9.68}$ | OOT | $44.90_{\pm0.88}$ | $45.37_{\pm0.57}$ |
| | 3 | GIN | $19.63_{\pm4.21}$ | $43.72_{\pm3.26}$ | $24.25_{\pm14.43}$ | $20.97_{\pm6.64}$ | OOT | $45.04_{\pm1.94}$ | |

## 4.4 CONDENSATION EFFICIENCY

Table. 6 presents the time consumed by BONSAI and other baselines. We note that while BONSAI is CPU-bound, all other algorithms are GPU-bound. Despite being CPU-bound, BONSAI, on average, is more than 7-times faster than the fastest baseline EXGC. In addition, all baselines require training on the full dataset. This algorithm design shows up in the running times where the condensation time is higher than the full dataset training time, negating the very purpose of graph condensation. More worryingly, the condensation process of all algorithms, except BONSAI, have higher carbon emissions than training on the full dataset (See Table 8 in Appendix).

Table 6: Condensation times of various methods in seconds at 0.5%. Condensation times that are higher than training on the full dataset itself are highlighted in red. The fastest condensation time for each dataset is highlighted in green.

| Dataset | GCOND | GDEM | GCSR | EXGC | GC-SNTK | GEOM | BONSAI | Full |
|---|---|---|---|---|---|---|---|---|
| Cora | 2738 | 105 | 5260 | 34.87 | 82 | 12996 | 2.60 | 24.97 |
| Citeseer | 2712 | 167 | 6636 | 34.51 | 124 | 15763 | 2.75 | 24.87 |
| Pubmed | 2567 | 530 | 1319 | 114.96 | 117 | OOT | 24.24 | 51.06 |
| Flickr | 1935 | 3405 | 17445 | 243.28 | 612 | OOT | 118.23 | 180.08 |
| Ogbn-arxiv | 14474 | 569 | OOT | 1594.83 | 12218 | OOT | 298.64 | 524.67 |
| Reddit | 30112 | 20098 | OOT | 6903.47 | 29211 | OOT | 1170.64 | 2425.68 |

**Component-wise analysis:** We discuss the individual time consumption by each component of BONSAI in App. B.4.1.

## 4.5 ABLATION STUDY AND IMPACT OF PARAMETERS

Fig. 4 presents an ablation study comparing BONSAI against three variants: **(i)** BONSAI-PPR, which omits the use of Personalized PageRank, **(ii)** BONSAI-Rev-$k$-NN, which substitutes the Rev-$k$-NN-based coverage maximization with random exemplar selection, but performs PPR, **(iii)** and random that does not do either PPR or Rev-$k$-NN. Before analyzing the trends, we emphasize an important distinction between BONSAI-Rev-$k$-NN and Random. While in BONSAI-Rev-$k$-NN, we randomly add computation trees till the condensation budget is exhausted, in Random, we iteratively add random nodes, till their induced subgraph exhausts the budget. Consequently, Random covers more diverse nodes with partial local neighborhood information, while BONSAI-Rev-$k$-NN selects a smaller node set with complete $L$-hop topology, enabling precise GNN embedding computation.

Several important insights emerge from this experiment. Random performs significantly worse than BONSAI, demonstrating that the condensed graph's information content substantially exceeds that of an equally sized subgraph induced by random node selection. This finding, consistent with the trend observed in Table 4, underscores the effectiveness of BONSAI over random sampling.

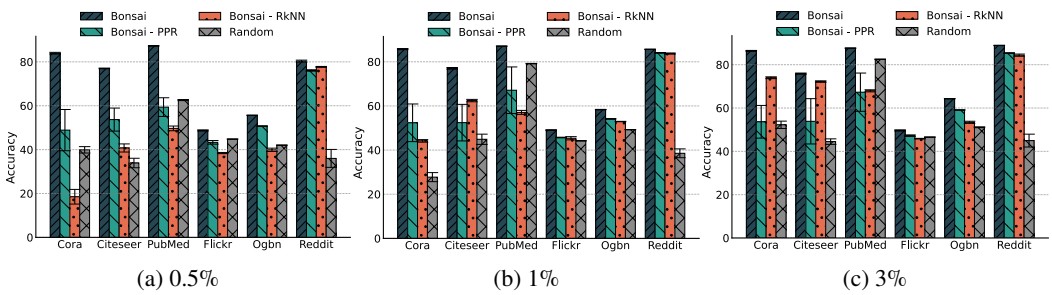

Figure 4: Ablation study of BONSAI.

Second, removal of either Rev-$k$-NN or PPR result in significant drops across most datasets, indicating both play important and complementary roles. While Rev-$k$-NN identifies the exemplars that are important, PPR sparsifies the graph making space for more exemplars to be added. Interestingly, the relative impact of these components varies with dataset compression and size. At higher compression rates (such as 0.5%), removing Rev-$k$-NN leads to higher drop in accuracy across most datasets. However, this trend reverses as the condensed dataset size increases. Closer examination reveals that higher-ranked exemplars typically have higher degrees in their $L$-hop neighborhoods. When the memory budget is constrained, high-ranked exemplars are often skipped because their $L$-hop neighborhoods exceed the available memory. This limitation diminishes as the budget increases. Consequently, at higher budgets, edge density increases, making graph sparsification (via PPR) increasingly critical. This explains why PPR becomes more important at higher budgets or in inherently dense graphs like Reddit.

Finally, we discuss the performance comparison of Random with BONSAI-Rev-$k$-NN. This is an interesting comparison since both represent two distinct mechanisms of random selection. In most cases, BONSAI-Rev-$k$-NN achieves a higher accuracy indicating that obtaining full $L$-hop topological information for a smaller set of nodes leads to better results than partial $L$-hop information over a broader set of nodes.

**Impact of Parameters:** We analyze the impact of sampling size and $k$ on Rev-$k$-NN in App. B.4.2.

## 5 CONCLUSIONS AND FUTURE WORKS

In this work, we have developed BONSAI, a novel graph condensation method that addresses critical limitations in existing approaches. By leveraging the fundamental role of computation trees in message-passing GNNs, BONSAI achieves superior performance in node classification tasks across multiple real-world datasets. Our method stands out as the first linear-time, model-agnostic graph condensation algorithm, offering significant improvements in both accuracy and computational efficiency. BONSAI's unique approach of encoding exemplar trees that maximize representation of the full training set's computation trees has proven to be highly effective. This strategy not only overcomes the paradoxical requirement of training on full datasets for condensation but also eliminates the need for repeated condensation when changing hyper-parameters or GNN architectures. Furthermore, BONSAI achieves substantial size reduction without resorting to fully-connected, edge-weighted graphs, thereby reducing computational demands. In contrast to baselines, BONSAI is completely CPU-bound, resulting in at least 17-times lower carbon emissions, making it a more environmentally friendly option. These features collectively position BONSAI as a significant advancement in the field of graph condensation, offering both performance benefits and sustainability advantages.

**Limitations and Future Works:** While existing research on graph condensation has primarily focused on node and graph classification tasks, GNNs have demonstrated their versatility across a broader spectrum of applications. In our future work, we aim to expand the scope of graph condensation by developing task-agnostic data condensation algorithms.

## REPRODUCIBILITY STATEMENT

To support the reproducibility of our work, we provide several resources in the paper and its supplementary materials. The source code of our models and algorithms is available at https://github.com/idea-iitd/Bonsai, which also includes details of how to train the baseline models in different settings. All theoretical results and assumptions are detailed in the Appendix A, ensuring clarity in our claims. Full experimental settings are documented in Appendix B.

ACKNOWLEDGEMENTS

We acknowledge the Yardi School of AI, IIT Delhi for supporting this research. This work was partially supported by the CSE Research Acceleration Fund of IIT Delhi. Sayan Ranu acknowledges the Nick McKeown Chair position endowment. Hariprasad Kodamana acknowledges IIT Delhi Abu Dhabi for partial support of this research. Samyak Jain acknowledges the generous grant received from Microsoft Research India to sponsor his travel to ICLR 2025; and Hudson River Trading LLC for funding his work on the project.

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

## A APPENDIX

### A.1 PROOF. OF THEOREM 1.

Recall the notation that $\mathbb{S} \subseteq \mathbb{T}$ denotes the subset of trees from which the representative power of all trees in $\mathbb{T}$ is being estimated. Let $|\mathbb{S}| = z$.

Corresponding to each $\mathcal{T}_u^L \in \mathbb{S}$, let us define an indicator random variable $X_u^v$ denoting whether $\mathcal{T}_v^L \in k\text{-NN}(\mathcal{T}_u^L)$. Therefore, we have:

$$X^v = \sum_{\forall \mathcal{T}_u^L \in \mathbb{S}} X_u^v = z\widetilde{\Pi}\left(\mathcal{T}_v^L\right) \tag{8}$$

Since each tree in $\mathbb{S}$ is sampled uniformly at random from $\mathbb{T}$, we have $E(X^v) = z\Pi\left(\mathcal{T}_v^L\right)$. We also note that $X^v \sim Binomial(z, \Pi\left(\mathcal{T}_v^L\right))$. Given any $\epsilon \geq 0$, from Chernoff bounds (Motwani & Raghavan, 1995), we have:

Upper tail bound:

$$P\left(X^v \geq (1+\epsilon)z\Pi\left(\mathcal{T}_v^L\right)\right) \leq \exp\left(-\frac{\epsilon^2}{2+\epsilon}z\Pi\left(\mathcal{T}_v^L\right)\right) \tag{9}$$

Lower tail bound:

$$P\left(X^v \leq (1-\epsilon)z\Pi\left(\mathcal{T}_v^L\right)\right) \leq \exp\left(-\frac{\epsilon^2}{2}z\Pi\left(\mathcal{T}_v^L\right)\right) \tag{10}$$

Combining the upper and tail bounds, we obtain:

$$P\left(\left|X^v - z\Pi\left(\mathcal{T}_v^L\right)\right| \geq \epsilon z\Pi\left(\mathcal{T}_v^L\right)\right) \leq 2\exp\left(-\frac{\epsilon^2}{2+\epsilon}z\Pi\left(\mathcal{T}_v^L\right)\right) \tag{11}$$

$$\Leftrightarrow P\left(\left|z\widetilde{\Pi}(\mathcal{T}_v^L) - z\Pi\left(\mathcal{T}_v^L\right)\right| \geq \epsilon z\Pi\left(\mathcal{T}_v^L\right)\right) \leq 2\exp\left(-\frac{\epsilon^2}{2+\epsilon}z\Pi\left(\mathcal{T}_v^L\right)\right) \tag{12}$$

$$\Leftrightarrow P\left(\left|\widetilde{\Pi}(\mathcal{T}_v^L) - \Pi\left(\mathcal{T}_v^L\right)\right| \geq \epsilon\Pi\left(\mathcal{T}_v^L\right)\right) \leq 2\exp\left(-\frac{\epsilon^2}{2+\epsilon}z\Pi\left(\mathcal{T}_v^L\right)\right) \tag{13}$$

Plugging $\theta$ from Eq. 5 in Eq. 13, we obtain $\theta = \epsilon\Pi\left(\mathcal{T}_v^L\right)$. Hence,

$$P\left(\left|\widetilde{\Pi}(\mathcal{T}_v^L) - \Pi\left(\mathcal{T}_v^L\right)\right| \geq \theta\right) \leq 2\exp\left(-\frac{\left(\frac{\theta}{\Pi(\mathcal{T}_v^L)}\right)^2}{2 + \frac{\theta}{\Pi(\mathcal{T}_v^L)}}z\Pi\left(\mathcal{T}_v^L\right)\right) \tag{14}$$

$$\leq 2\exp\left(-\frac{\theta^2}{2\Pi\left(\mathcal{T}_v^L\right)+\theta}z\right) \tag{15}$$

For any $\mathcal{T}_v^L \in \mathbb{T}$, $\Pi\left(\mathcal{T}_v^L\right) \leq 1$. Thus,

$$P\left(\left|\widetilde{\Pi}(\mathcal{T}_v^L) - \Pi\left(\mathcal{T}_v^L\right)\right| \geq \theta\right) \leq 2\exp\left(-\frac{\theta^2}{2+\theta}z\right) \tag{16}$$

Now, if we want a confidence interval of $1 - \delta$ (as stated in Eq. 5), we have:

$$\delta \geq 2 \exp\left(-\frac{\theta^2}{2+\theta}z\right) \tag{17}$$

$$\Leftrightarrow \frac{\delta}{2} \geq \exp\left(-\frac{\theta^2}{2+\theta}z\right) \tag{18}$$

$$\Leftrightarrow \ln\frac{\delta}{2} \geq -\frac{\theta^2}{2+\theta}z \tag{19}$$

$$\Leftrightarrow \ln\frac{2}{\delta} \leq \frac{\theta^2}{2+\theta}z \tag{20}$$

$$\Leftrightarrow \ln\frac{2}{\delta} \leq \frac{\theta^2}{2+\theta}z \tag{21}$$

$$\Leftrightarrow z \geq \frac{\ln\left(\frac{2}{\delta}\right)(2+\theta)}{\theta^2} \tag{22}$$

Hence, if we sample at least $\frac{\ln\left(\frac{2}{\delta}\right)(2+\theta)}{\theta^2}$ trees in $\mathbb{S}$, then for any $\mathcal{T}_v^L \in \mathbb{T}$:

$$\widetilde{\Pi}(\mathcal{T}_v^L) \in [\Pi\left(\mathcal{T}_v^L\right) - \theta, \Pi\left(\mathcal{T}_v^L\right) + \theta] \text{ with probability at least } 1 - \delta. \qquad \square \tag{23}$$

## A.2 NP-HARDNESS: PROOF OF THEOREM 1

PROOF. To establish NP-hardness of our proposed problem we reduce it to the classical *Set Cover* problem (Cormen et al., 2009).

**Definition 7** (Set Cover). *Given a budget $b$ and a collection of subsets $\mathcal{S} = \{S_1, \cdots, S_m\}$ from a universe of items $U = \{u_1, \cdots, u_n\}$, i.e., $\forall S_i \in \mathcal{S}$, $S_i \subseteq U$, the Set Cover problem seeks to determine whether there exists a subset $\mathcal{S}' \subseteq \mathcal{S}$ such that $|\mathcal{S}'| = b$ and it covers all items in the universe, i.e., $|\bigcup_{\forall S_i \in \mathcal{S}'} S_i| = U$.*

We show that given any instance of a Set Cover problem $\langle \mathcal{S}, U \rangle, b$, it can be mapped to the problem of maximizing the representative power (Eq. 7).

Given an instance of the Set Cover problem, we construct a database of computation trees $\mathbb{T} = \mathbb{T}_U \cup \mathbb{T}_{\mathcal{S}}$. For each $u_j \in U$, we add a computation tree $\mathcal{T}_{u_j} \in \mathbb{T}_U$. For each $S_i \in \mathcal{S}$, we add a tree $\mathcal{T}_{S_i} \in \mathbb{T}_{\mathcal{S}}$. If an item $u_j \in S_i$, then we have $\mathcal{T}_{u_j} \in \text{Rev-}k\text{-NN}(\mathcal{T}_{S_i})$ (equivalently $\mathcal{T}_{S_i} \in k\text{-NN}(\mathcal{T}_{u_j})$).

With this construction, we can state that there exists a Set Cover of size $b$ if and only if there exists a solution set $\mathbb{A} \subseteq \mathbb{T}$ such that $\Pi(\mathbb{A}) = \frac{|\mathbb{T}_U|}{|\mathbb{T}|}$. $\qquad \square$

## A.3 PROOFS OF MONOTONICITY, SUBMODULARITY

**Theorem 3** (Monotonicity). $\forall \mathbb{A}' \supseteq \mathbb{A}, \Pi(\mathbb{A}') - \Pi(\mathbb{A}) \geq 0$, *where $\mathbb{A}$ and $\mathbb{A}'$ are sets of computation trees.*

*Proof.* Since the denominator in $\Pi(\mathbb{A})$ is constant, it suffices to prove that the numerator is monotonic. This reduces to establishing that:

$$\left| \bigcup_{\forall \mathcal{T}_v^L \in \mathbb{A}'} \text{Rev-}k\text{-NN}\left(\mathcal{T}_v^L\right) \right| \geq \left| \bigcup_{\forall \mathcal{T}_v^L \in \mathbb{A}} \text{Rev-}k\text{-NN}\left(\mathcal{T}_v^L\right) \right| \tag{24}$$

This inequality holds true because the union operation is a monotonic function. As $\mathbb{A}'$ is a superset of $\mathbb{A}$, the union over $\mathbb{A}'$ will always include at least as many elements as the union over $\mathbb{A}$. $\qquad \square$

**Theorem 4** (Submodularity). $\forall \mathbb{A}' \supseteq \mathbb{A}, \Pi(\mathbb{A}' \cup \{\mathcal{T}\}) - \Pi(\mathbb{A}') \leq \Pi(\mathbb{A} \cup \{\mathcal{T}\}) - \Pi(\mathbb{A})$, *where $\mathbb{A}$ and $\mathbb{A}'$ are sets of computation trees and $\mathcal{T}$ is a computation tree.*

PROOF BY CONTRADICTION. Let us assume

$$\exists \mathbb{A}' \supseteq \mathbb{A}, \Pi(\mathbb{A}' \cup \{\mathcal{T}\}) - \Pi(\mathbb{A}') > \Pi(\mathbb{A} \cup \{\mathcal{T}\}) - \Pi(\mathbb{A}) \tag{25}$$

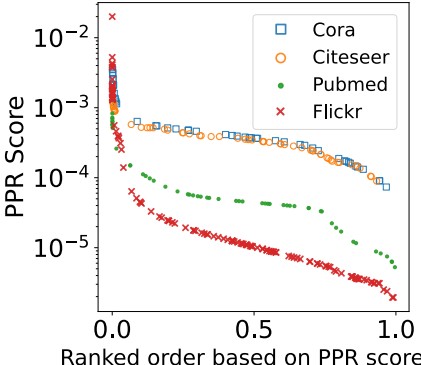

Figure 5: Distribution of PPR scores against node ranks.

Eq. 25 implies:

$$\text{Rev-}k\text{-NN}(\mathcal{T}) \setminus \bigcup_{\forall \mathcal{T}' \in \mathbb{A}'} \text{Rev-}k\text{-NN}(\mathcal{T}') \supseteq \text{Rev-}k\text{-NN}(\mathcal{T}) \setminus \bigcup_{\forall \mathcal{T}' \in \mathbb{A}} \text{Rev-}k\text{-NN}(\mathcal{T}') \qquad (26)$$

$$\implies \mathbb{A} \supseteq \mathbb{A}', \text{ which is a contradiction.} \qquad \qquad \square$$

### A.4    Sparsification through Personalized PageRank (PPR)

Let $\mathcal{V}_S$ be the node set in the initial condensed graph, which can be partitioned into two disjoint subsets: $\mathcal{V}_{root}$ and $\mathcal{V}_{ego}$, where $\mathcal{V}_{root} = \{v \mid \exists \mathcal{T}_v^L \in \mathbb{A}\}$, the *root nodes*, that serve as the root node of some exemplar in $\mathbb{A}_{greedy}$, and $\mathcal{V}_{ego} = \mathcal{V}_S \setminus \mathcal{V}_{root}$, the *ego nodes*, that are included because they fall within the $L$-hop neighborhood of a root node. We propose to employ *Personalized PageRank (PPR)* (Page et al., 1998) with teleportation only to root nodes to identify and prune ego nodes with minimal impact on the root node embeddings. Starting with timestamp $t = 0$, we proceed as follows:

1. Compute the PPR distribution $\boldsymbol{\pi}_t$ for $t \geq 0$ defined by:

$$\boldsymbol{\pi}_t = (1 - \beta)\mathbf{A}\boldsymbol{\pi}_{t-1} + \beta\mathbf{e} \qquad (27)$$

   where: $\boldsymbol{\pi}_0[i] = \frac{1}{|\mathcal{V}_S|}$ for all $i$, $\mathbf{e}[i] = \frac{1}{|\mathcal{V}_{root}|}$ if $v_i \in \mathcal{V}_{root}$, otherwise 0, $\mathbf{A}$ is the normalized adjacency matrix, $\beta$ is the teleportation probability.
2. Obtain the PPR vector $\boldsymbol{\pi}$ after convergence where $\boldsymbol{\pi}_t = \boldsymbol{\pi}_{t-1}$.
3. Sort the nodes in $\boldsymbol{\pi}$ by their PPR scores in descending order such that $\forall i, \boldsymbol{\pi}[i-1] \geq \boldsymbol{\pi}[i]$.
4. Define the knee point $i_{knee}$ as:

$$i_{knee} = \arg\max_i \{\boldsymbol{\pi}[i+1] + \boldsymbol{\pi}[i-1] - 2 \cdot \boldsymbol{\pi}[i]\} \qquad (28)$$

5. Prune all ego nodes with PPR scores below $\boldsymbol{\pi}[i_{knee}]$.
6. Use the freed-up space to include additional exemplar trees using Algorithm 1.
7. Repeat steps 1–6 iteratively until $i_{knee} \geq \theta$, where $\theta$ is a threshold setting the minimum number of nodes to be removed.

The rationale for this pruning process can be outlined as follows. The PPR distribution $\boldsymbol{\pi}_t$ iteratively updates based on the transition matrix $\mathbf{A}$ and teleportation vector $\mathbf{e}$. Given that $\boldsymbol{\pi}_0$ is initialized uniformly and $\mathbf{e}$ assigns higher probabilities to root nodes, $\boldsymbol{\pi}_t$ converges to a distribution where nodes more connected to root nodes have higher scores (See Fig. 5 in Appendix). After convergence, the vector $\boldsymbol{\pi}$ represents the steady-state probabilities of nodes in the graph. Sorting $\boldsymbol{\pi}$ by descending scores ensures that nodes with the highest influence on root nodes are prioritized. The knee point $i_{knee}$ is identified as the maximum curvature in the sorted PPR scores. This point is where the rate of change in scores shifts most significantly, indicating a transition between nodes of high and low influence. Nodes with PPR scores below $\boldsymbol{\pi}[i_{knee}]$ have minimal impact on root node embeddings and removing these nodes and edges incident on them effectively sparsifies the graph without substantial loss of information. Additional exemplar trees can then be incorporated into the available space using the greedy algorithm as given by Algorithm 1.

### A.5 Complexity Analysis

For this analysis, we assume $|\mathcal{V}| = n$ and $|\mathcal{E}| = m$. In sync with Fig. 3, the computation structure of BONSAI can be decomposed into four components.

1. **Embed trees into WL-space:** This operation requires a pass through each twice, and is identical to the message-pass structure of a GNN. Hence, the computation cost is bounded to $\mathcal{O}(m)$. A crucial distinction from a GNN is that it does not require back-propagation and hence is fully CPU-bound.
2. **Rev-$k$-NN:** As discussed in § 3.1.1, computing Rev-$k$-NN consumes $\mathcal{O}(n)$ time with sampling.
3. **Coverage Maximization:** Each iteration in Alg. 1, iterates over all nodes (trees) in the graph ($\mathbb{T}$) that have not yet been added to the set of exemplars. For each node (equivalently, tree rooted at this node), we compute its marginal gain (line 3 in Alg. 1), which is bounded by the sample size $z$, since the cardinality of a Rev-$k$-NN set is bounded by the number of trees sampled for Rev-$k$-NN computation ((§ 3.1.1)). Since $z \ll n$, the complexity per iteration is $\mathcal{O}(zn) \approx \mathcal{O}(n)$. The number of iterations is bounded by the size of the condensation budget, which we typically set to less than 3% of the full dataset and hence has negligible impact on the complexity.
4. **PPR:** PPR can be computed in linear time (Lofgren et al., 2014). Since the size of the condensed set is at most the full graph, this bounds the cost of PPT to $\mathcal{O}(n)$.

Combining all these components, the cost of BONSAI is bounded by $\mathcal{O}(n + m)$.

## B Experimental Setup

Across all datasets, except MAG240M, we maintain a train-validation-test split ratio of $60 : 20 : 20$. In MAG240M, we use a ratio of 80:10:10.

### B.1 Hardware Configuration

All experiments were conducted on a high-performance computing system with the following specifications:

- **CPU**: 96 logical cores
- **RAM**: 512 GB
- **GPU**: NVIDIA A100-PCIE-40GB

### B.2 Software Configuration

The software environment for our experiments was configured as follows:

- **Operating System**: Linux (Ubuntu 20.04.4 LTS (GNU/Linux 5.4.0-124-generic x86_64)) )
- **PyTorch Version**: 1.13.1+cu117
- **CUDA Version**: 11.7
- **PyTorch Geometric Version**: 2.3.1

### B.3 Additional Experimental Parameters

- **Number of layers in evaluation models**: 2 (with RELU in between) for GCN, GAT, and GIN. The MLP used in GIN is a simple linear transform with a bias defined by the following equation $\mathbf{WX} + \mathbf{b}$ where $\mathbf{X}$ is the input design matrix.
- **Value of $k$ in Rev-$k$-NN**: 5
- **Hyper-parameters Baselines:** We use the config files shared by the authors. We note that the benchmark datasets are common between our experiments and those used in the baselines.

### B.4 Additional Experiments

### B.4.1 Running time analysis of BONSAI

Fig. 7 presents the time consumed by the various components within BONSAI. We observe that most of the time is spent in PPR computation, except in Reddit, where Rev-$k$-NN is a dominant contributor.

Table 7: Component-wise analysis of BONSAI's running times in seconds.

| Dataset | WL | Rev-$k$-NN | Greedy | PPR | Full |
|---|---|---|---|---|---|
| Cora | 0.87 | 0.05 | 0.005 | 1.68 | 2.60 |
| Citeseer | 0.97 | 0.07 | 0.005 | 1.71 | 2.75 |
| Pubmed | 8.89 | 1.39 | 0.030 | 13.93 | 24.24 |
| Reddit | 682.80 | 293.15 | 0.696 | 193.99 | 1170.64 |

This trend can be explained by the high density of Reddit, where the average degree is 100 in contrast to $< 5$ in the other networks.

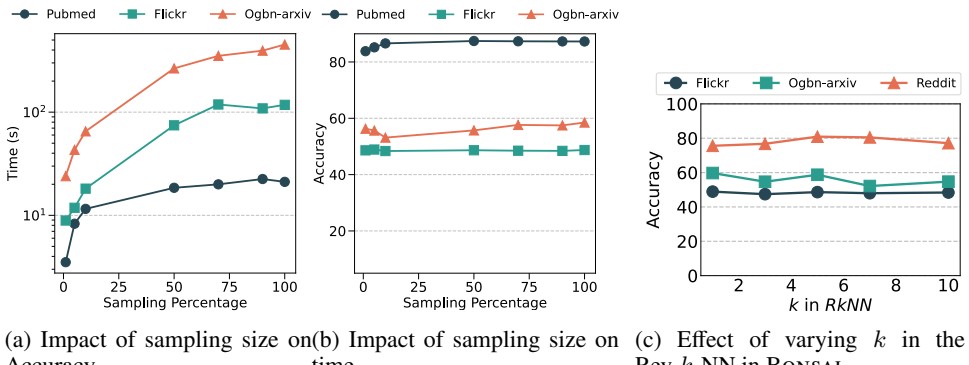

(a) Impact of sampling size on Accuracy
(b) Impact of sampling size on time
(c) Effect of varying $k$ in the Rev-$k$-NN in BONSAI.

Figure 6: Impact of $k$ in Rev-$k$-NN and sampling size for Rev-$k$-NN approximation on the performance of BONSAI.

### B.4.2 IMPACT OF PARAMETERS ON BONSAI

BONSAI has two parameters, both related to the computation of Rev-$k$-NN. The value of $k$ in Rev-$k$-NN and the sample size to accurately approximate the representative power.

Fig. 6a and Fig. 6b present the impact of sample size on accuracy and time, respectively. As expected, the general trend shows that increasing the sample size leads to higher accuracy and longer running times. However, we note that the increase in accuracy is minor. This observation is consistent with Lemma 1, which states that a small sample size is sufficient to achieve a high-quality approximation.

Fig. 6c presents the impact of $k$ on accuracy. While the performance is generally stable against $k$, the peak accuracy is typically achieved when $k$ is set to 5.

### B.5 ADDITIONAL TRAINING TIMES

Fig. 7 presents the training times on condensed datasets in Cora, Citeseer, Pubmed, and Reddit at 0.5% size. Figs. 8 and 9 presents the training times on condensed datasets in Cora, Citeseer, Pubmed, Flickr, Ogbn-arxiv, and Reddit at 1%, and 3% sizes. BONSAI is faster on all architectures. The gap between BONSAI and the baselines increase with increase in condensed dataset size. This is more pronounced in larger datasets like Flickr, Ogbn-arxiv, and Reddit because the number of edges is $O(|V|^2)$ for the baselines and forward pass through GNNs is usually $O(|E|)$. Additionally, the condensed dataset created by GCOND and GDEM for Reddit at 3% size cannot be used to train basic PyTorch Geometric models for GAT and GIN, resulting in out-of-memory (OOM) errors because the backward pass graph maintained by PyTorch is larger.

### B.6 CARBON EMISSIONS

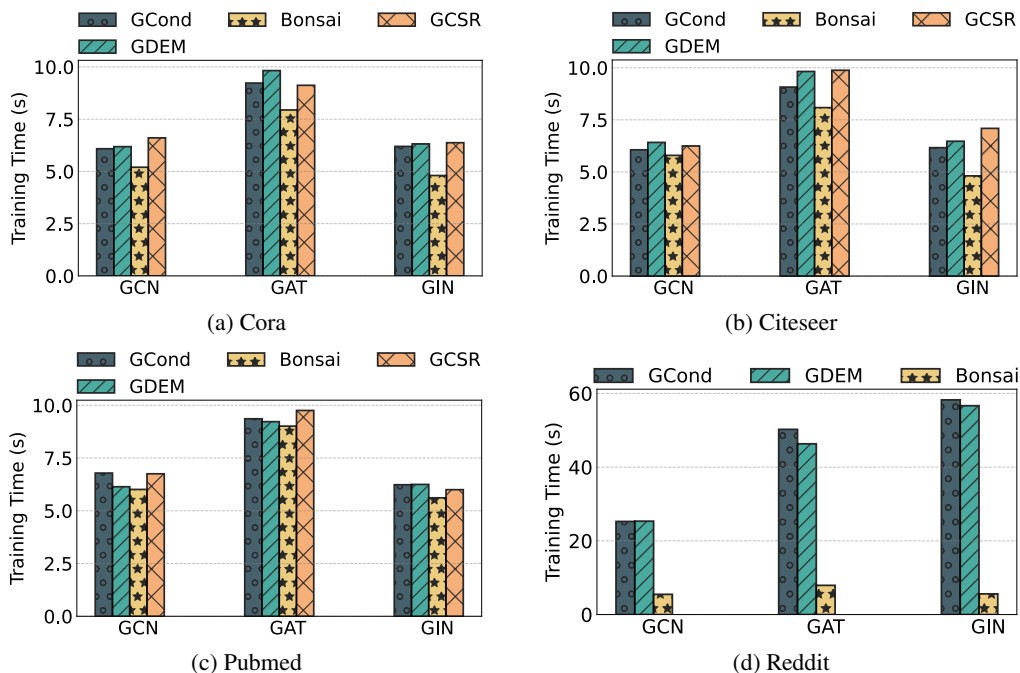

Figure 7: Comparing time required to train models on datasets condensed by GCOND, GDEM, BON-SAI, and GCSR at 0.5% of Cora, Citeseer, Pubmed, and Reddit. Note that there is no condensed dataset for Reddit output by GCSR due to OOT.

Table 8: Estimated $CO_2$ emissions from condensation of various methods in seconds at 0.5%. Emissions that are higher than training on the full dataset itself are highlighted in red. The least emission from condensation for each dataset is highlighted in green. $CO_2$ emissions are computed as 10.8kg per 100 hours for Nvidia A100 GPU and 4.32kg per 100 hours for 10 CPUs of Intel Xeon Gold 6248 (Lacoste et al., 2019).

| Dataset | GCOND | GDEM | GCSR | EXGC | GC-SNTK | GEOM | BONSAI | Full |
|---------|-------|------|------|------|---------|------|--------|------|
| Cora | 82.14 | 3.15 | 157.80 | 1.05 | 2.46 | 389.88 | 0.03 | 0.75 |
| Citeseer | 81.36 | 5.01 | 199.08 | 1.03 | 3.72 | 472.89 | 0.03 | 0.75 |
| Pubmed | 77.01 | 15.90 | 39.57 | 3.45 | 3.51 | ≥540.00 | 0.3 | 1.53 |
| Flickr | 58.05 | 102.15 | 523.35 | 7.30 | 18.36 | ≥540.00 | 1.42 | 5.40 |
| Ogbn-arxiv | 434.22 | 17.07 | ≥540.00 | 46.49 | 366.54 | ≥540.00 | 4.18 | 15.74 |
| Reddit | 903.36 | 602.94 | ≥540.00 | 207.10 | 876.33 | ≥540.00 | 17.34 | 72.77 |

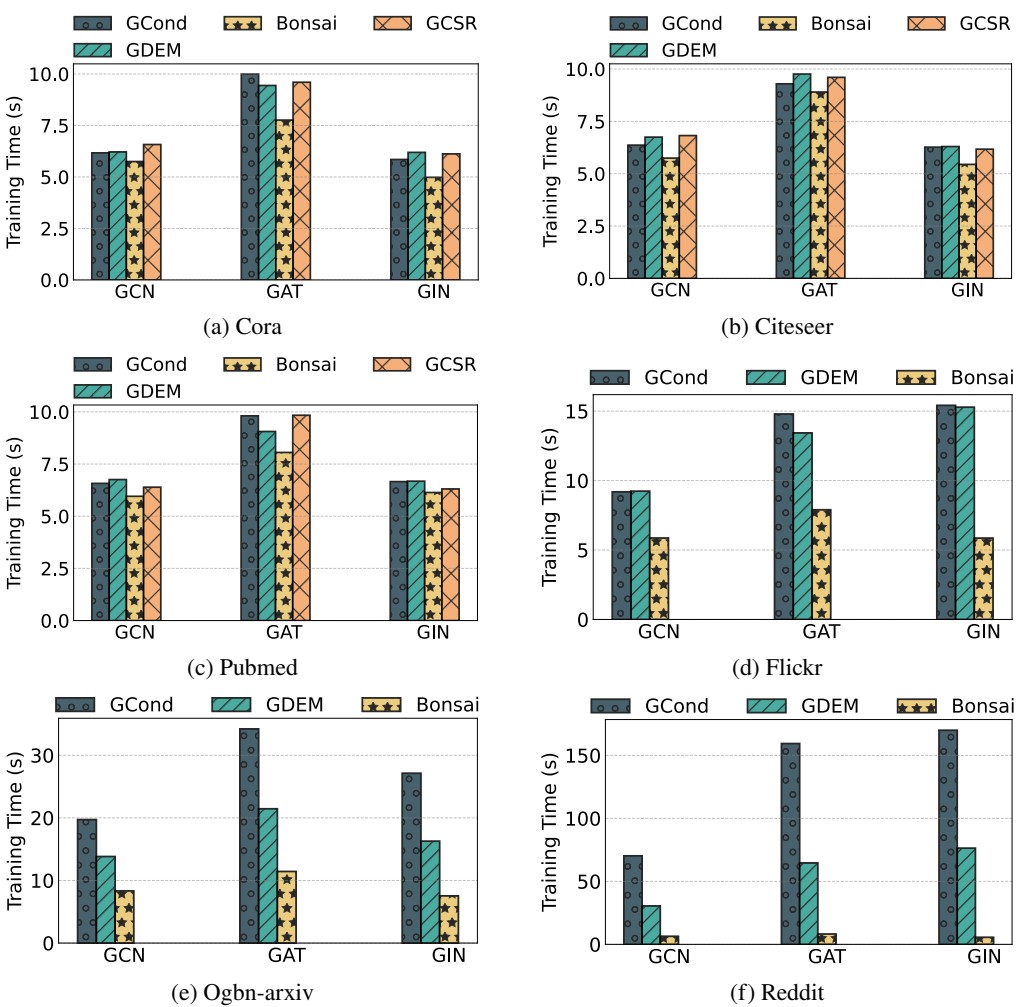

Figure 8: Comparing time required to train models on datasets condensed by GCOND, GDEM, BON-SAI, and GCSR at 1% of Cora, Citeseer, Pubmed, Flickr, Ogbn-arxiv, and Reddit. Note that there is no condensed dataset for Flickr, Ogbn-arxiv, Reddit output by GCSR due to OOT.

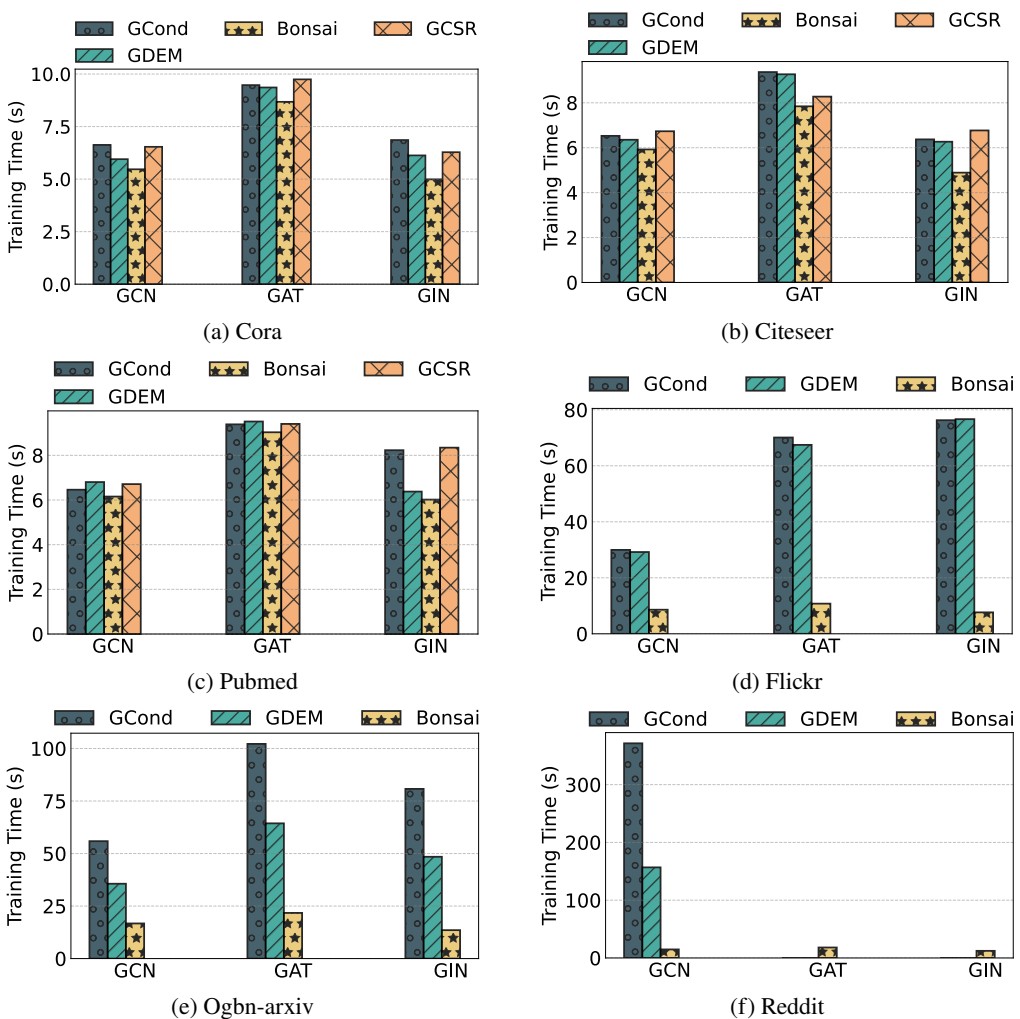

Figure 9: Comparing time required to train models on datasets condensed by GCOND, GDEM, BON-SAI, and GCSR at 3% of Cora, Citeseer, Pubmed, Flickr, Ogbn-arxiv, and Reddit. Note that there is no condensed dataset for Flickr, Ogbn-arxiv, Reddit output by GCSR due to OOT; and condensed datasets produced by GCOND and GDEM for Reddit cannot train on standard 2-layered PyTorch Geometric GAT and GIN due to OOM.

