# OpenReview forum: "Bonsai: Gradient-free Graph Condensation for Node Classification"
_ICLR.cc/2025/Conference — ICLR 2025 Poster_

### Official Review · Reviewer_q9ty · 2024-10-28

**Soundness:** 3
**Presentation:** 3
**Contribution:** 2
**Rating:** 6
**Confidence:** 3

**Summary:**

This paper proposes Bonsai, a linear-time, model-agnostic graph distillation algorithm for node classification. It observe three limitations of previous works, Full Gnn training, Distilling to a fully-connected graph and Model-specific. To address these limitations, Bonsai  aims to identify a set of b exemplar computation trees that optimally represent the full training set.

**Strengths:**

S1: The problem is important in this field.

S2: Presentation is good. Good model name.

S3: The solution looks novel to me.

**Weaknesses:**

W1: The experiments are not very standard.

W2: Some recent works are not included as baseline.

**Questions:**

Q1: Why the authors choose 0.5%, 1% and 3% as $S_r$? This setting does not align with previous works.

Q2: Recently, a lot of works have studied the efficiency of graph condensation, e.g., Exgc [1] and GC-SNTK [2]. These two methods should be included as baselines when comparing condensation time. By the way, it would be better to present time comparison via table than figure.

Q3: Other graph condensation methods should be included for accuracy comparison, e.g., SGDD [3], SFGC [4], GEOM [5].

[1] Exgc: Bridging efficiency and explainability in graph condensation

[2] Fast Graph Condensation with Structure-based Neural Tangent

[3] Does graph distillation see like vision dataset counterpart

[4] Structure-free graph condensation: From large-scale graphs to condensed graph-free data

[5] Navigating Complexity: Toward Lossless Graph Condensation via Expanding Window Matching

---

> ### Author Response · Authors · 2024-11-19
> **Response to Reviewer q9ty- Part 1**
>
> We thank the reviewer for the positive feedback and constructive comments on our work. Please find below  our clarifications and changes made in the revised manuscript to address the concerns raised.
>
> **W1 and Q1: Why the authors choose 0.5%, 1% and 3% as $S_r$? This setting does not align with previous works.**
>
> **Answer:** The compression ratios established in GCond have become a de facto standard, with subsequent works largely adopting the same approach. However, our analysis reveals significant issues with this methodology that necessitate a fresh examination.
>
>   * **Ensuring fairness and consistency:** The first concern is the inconsistency in compression ratios across datasets. For instance, while Cora uses ratios of 1.3%, 2.6%, and 5.2%, Reddit employs much smaller ratios of 0.05%, 0.1%, and 0.2%. Notably, there appears to be no documented justification for these widely varying ratios. **Our approach implements uniform compression ratios across all datasets, eliminating potential dataset-specific biases.**
>
>   * **Interpretability:** The lack of uniformity obscures important insights. As examples, in GCond and GDEM, the **compression ratios of 5.2% and 3.6% in Cora and Citeseer actually translate to using all nodes in the training set** as these percentages match the proportion of nodes in the training data. More critically, as we point out in `Table 2`, this results in **distilled datasets being larger than the full dataset!** Thus, in practice, it means no compression with respect to node count, and inflation with respect to edge counts.
>
> By standardizing compression ratios across datasets, we provide more transparent evaluation, clearer understanding of true compression effectiveness, and fair comparison across different datasets.
>
> **W2 and Q2: Recently, a lot of works have studied the efficiency of graph condensation, e.g., Exgc [1] and GC-SNTK [2]. These two methods should be included as baselines when comparing condensation time. By the way, it would be better to present time comparison via table than figure.**
>
> **Answer:** We have included comparison to both Exgc and GC-SNTK in accuracy (`Table 5`) and distillation times (`Table 7`). We outperfom both across all datasets except in ogbg-arxiv at 0.5% and 1%.
>
> As suggested, we now present distillation times as a table (`Table 7`). The same table is also produced below for easy reference. In addition, to further showcase the benefit of Bonsai, we present the carbon emissions for distillation from each of the techniques (`Table 9`). As evident from the table below, Bonsai is **7 times faster** on average over EXGC, the 2nd fastest baseline, and **reduces carbon footprint by 17 times.**
>
> **Distillation times** (in seconds):
>
> | Dataset | Gcond | Gdem | Gcsr | Exgc | Gcsntk | Geom | Bonsai | Full training|
> | --- | --- | --- | --- | --- | --- | --- | --- | --- |
> | Cora | 2738 | 105 | 5260 | 34.87 | 82 | 12996 | **2.76** | 24.97 |
> | Citeseer | 2712 | 167 | 6636 | 34.51 | 124 | 15763 | **2.61** | 24.87 |
> | Pubmed | 2567 | 530 | 1319 | 114.96 | 117 | OOT | **24.84** | 51.06 |
> | Flickr | 1935 | 3405 | 17445 | 243.28 | 612 | OOT | **118.23** | 180.08 |
> | Ogbn-arxiv | 14474 | 569 | OOT | 1594.83 | 12218 | OOT | **348.24** | 524.67 |
> | Reddit | 30112 | 20098 | OOT | 6903.47 | 29211 | OOT | **1445.00** | 2425.68 |
>
> **Carbon emissions**: Estimated CO$_2$ emissions from distillation of various methods in seconds at 0.5\% compression ratio. CO$_2$ emissions are computed as 10.8kg per 100 hours for Nvidia A100 GPU and 4.32kg per 100 hours for 10 CPUs of Intel Xeon Gold 6248[1].
>
> | Dataset | Gcond | Gdem | Gcsr | Exgc | Gcsntk | Geom | Bonsai | Full training|
> | --- | --- | --- | --- | --- | --- | --- | --- | --- |
> | Cora | 82.14 | 3.15 | 157.80 | 1.05 | 2.46 | 389.88 | **0.03** | 0.75 |
> | Citeseer | 81.36 | 5.01 | 199.08 | 1.03 | 3.72 | 472.89 | **0.03** | 0.75 |
> | Pubmed | 77.01 | 15.90 | 39.57 | 3.45 | 3.51 | ≥540.00 | **0.3** | 1.53 |
> | Flickr | 58.05 | 102.15 | 523.35 | 7.30 | 18.36 | ≥540.00 | **1.42** | 5.40 |
> | Ogbn-arxiv | 434.22 | 17.07 | ≥540.00 | 46.49 | 366.54 | ≥540.00 | **4.18** | 15.74 |
> | Reddit | 903.36 | 602.94 | ≥540.00 | 207.10 | 876.33 | ≥540.00 | **17.34** | 72.77 |
>
> [1] Alexandre Lacoste, Alexandra Luccioni, Victor Schmidt, and Thomas Dandres. Quantifying the carbon emissions of machine learning.

---

> > ### Author Response · Authors · 2024-11-19
> > **Response to Reviewer q9ty- Part 1**
> >
> > **W2 and Q3: Other graph condensation methods should be included for accuracy comparison, e.g., SGDD [3], SFGC [4], GEOM [5].**
> >
> > **Answer:** We have added GEOM to `Table 5` as well as in our running time comparison. We outperform GEOM across all datasets, while being more than 50-times faster (`Table 7`) and 10000-times lower carbon emissions (`Table 9` in Appendix).
> >
> > Regarding SFGC, as we note in lines 407-410, we already include three algorithms, namely GDEM, GCSR and GEOM that have outperformed them in the literature. Second, SFGC requires training on the full dataset 200 times, which makes them impratical for graph distillation, where the goal is to avoid training on the full dataset in the first place. Bonsai and GDEM are the only two methods that achieves this goal.
> >
> > SGDD has also been outperformed in the literature by GCSR and GDEM. Additionally, in our experiments, it produced NaN repeatedly. This issue of non-reproducibility has been reported by multiple users in their github repo (https://github.com/RingBDStack/SGDD/issues/) and the authors have acknowledged a bug in the code with the promise to release an updated version, which have not been released till date.

---

> > > ### Comment · Reviewer_q9ty · 2024-11-22
> > >
> > > I thank the author for conducting so many additional experiments. I raise score. I don’t seem to have noticed the threshold for "out of time", perhaps I missed it. If the author hasn’t included it in the paper, I would recommend adding it.

---

> > > > ### Author Response · Authors · 2024-11-22
> > > > **thank you for raising score**
> > > >
> > > > We thank the reviewer for supporting our work and raising the score. The threshold of OOT (out of time) is mentioned in the caption of Table 5, which is 5 hours. We feel this is a reasonable time length since full training even on the largest dataset of Reddit finishes in 40 minutes (as shown in Table 7).
> > > >
> > > > Do not hesitate to let us know if there is anything more we can do to further convince you of the merits of our work.

---

### Official Review · Reviewer_n8k1 · 2024-11-01

**Soundness:** 3
**Presentation:** 3
**Contribution:** 2
**Rating:** 6
**Confidence:** 4

**Summary:**

The paper presents Bonsai, a gradient-free graph distillation method for node classification that overcomes limitations in existing approaches. By selecting exemplar computation trees rather than synthesizing fully connected graphs, Bonsai efficiently distills graphs without needing full dataset training or specific GNN architectures. This approach achieves higher accuracy and faster distillation across various datasets and GNN models.

**Strengths:**

1. The performance of bonsai is impressive, including the cross-arch results.
2. The proof of maximizing the representative power of exemplars is NP-hard is simple but attracting.

**Weaknesses:**

See questions.

**Questions:**

1. Could the authors clarify their specific contributions to Mirage [A]? While Mirage highlights the importance of the computation tree in graph dataset condensation, their primary focus is on the graph level (with node-level experiments included in the appendix). Thus, extending their approach to the node level by following the 1-WL test may not represent a substantial novelty.
2. Regarding Fig. 4(b), although the authors emphasize the significance of the RkNN and PPR components, the random exemplar selection still yields considerable results. How do the authors interpret this outcome? Could they also provide a performance comparison for $\mathbf{S_r}$ using different exemplars on datasets like Cora, Ogbn-arxiv, and Reddit? Including results for random selection would further clarify the comparison.
3. From my perspective, selecting datasets using a sampling strategy seems more akin to traditional graph reduction, sparsification, or coarsening methods, rather than directly aligning with the field of graph condensation. Therefore, it’s challenging to accept the significant improvements claimed over random or herding baselines. Could the authors provide an intuitive example to support their approach?

[A] Mridul Gupta, Sahil Manchanda, HARIPRASAD KODAMANA, and Sayan Ranu. Mirage: Model
agnostic graph distillation for graph classification. ICLR 2024.

---

> ### Author Response · Authors · 2024-11-19
> **Response to Reviewer n8k1 - Part 1**
>
> We thank the reviewer for their positive comments and constructive feedback. We have carefully addressed the concerns and incorporated the suggested revisions, as detailed below.
>
> **Q1. Could the authors clarify their specific contributions to Mirage [A]? While Mirage highlights the importance of the computation tree in graph dataset condensation, their primary focus is on the graph level (with node-level experiments included in the appendix). Thus, extending their approach to the node level by following the 1-WL test may not represent a substantial novelty.**
>
> **Answer:** We have updated the manuscript (`Sec 1.1 and App A.1`) with a detailed discussion on the novel contributions of Bonsai with respect to Mirage.
>
> The similarities between Mirage and Bonsai end at their shared goal of analyzing computation trees for distillation. **The fundamental limitation of Mirage lies in relying on tree isomorphisms, which makes them limited to graphs annotated with only single, discrete node labels** (such as atom-types in chemical compounds). Hence, **Mirage does not work on general purpose graphs** where nodes are annotated with high-dimensional feature vectors.
>
> Let us elaborate.
>
> **Summary of Mirage:** Given a graph database, Mirage identifies all unique computation trees of a certain depth in the graph database. Two trees are non-unique if they are isomorphic to each other. Next, each graph is represented as a set of computation trees, on which frequent itemset mining is performed. Trees in these frequent itemsets form the distilled dataset.
>
> **What breaks in our setting?** Tree isomorphism does not work when each node is attributed with high-dimensional feature vectors since treating feature vectors as labels means two trees are isomorphic if they are topologically identical and the **mapped nodes across the graphs are annotated with identical feature vectors.** In Bonsai, we make no such assumption.
>
> The algorithm design of Bonsai is entirely different. We make the following novel contributions:
>
> * **Reverse $k$-NN to rank tree importance:** Bonsai embeds computation trees into a feature space using WL-kernel and **ranks** each tree based on the density in its neighborhood (Reverse $k$-NN).
> * **Fast reverse $k$-NN through sampling:** Reverse $k$-NN is expensive ($O(n^2)$). Bonsai proposes an efficient sampling strategy with provable guarantees on the sample_size-approximation_error trade-off.
> * **Coverage maximization:** The core idea is to select a subset of trees that are representative of the entire set. Hence, we select trees located in dense neighborhoods and diverse from each other. This ensures that all trees that are not selected in distilled set are likely to have a close neighbor (in the embedding space) in the distilled set. This is achieved through coverage maximization (`Sec 3.2`).
> * **Theoretical guarantees:** We prove coverage maximization is NP-hard, monotonic and submodolar. Hence, greedy selection provides $1-1/e$ approximation guarantee.
> * **Sparsification** of the distilled dataset is performed through personalized page rank.
>
> **Empirical validation:** To further leave no room for ambiguity, we applied Mirage on Cora and Citeseer, and in both dataset all computation trees were unique leading to no compression.

---

> > ### Author Response · Authors · 2024-11-19
> > **Response to Reviewer n8k1 - Part 2**
> >
> > **Q2. Regarding Fig. 4(b), although the authors emphasize the significance of the RkNN and PPR components, the random exemplar selection still yields considerable results. How do the authors interpret this outcome? Could they also provide a performance comparison for using different exemplars on datasets like Cora, Ogbn-arxiv, and Reddit? Including results for random selection would further clarify the comparison.**
> >
> > **Answer:** We apologize for not being articulate enough in our analysis of the ablation study. To correct this, we have made the following changes to the updated manuscript (`Sec 4.5` to be specific).
> > * **Additional Experiments:** As suggested, we now perform ablation study **across all datasets and all compression ratios**. The results are available in updated `Figure 4`.
> > * **Added Random:** We have added Random to the ablation study. As evident from `Fig. 4`, across most datasets, Bonsai is dramatically better than Random.  FlickR is an exception, where Random does not outperform, but comes close to the performance of Bonsai. However, this trend has also been reported across all other baselines, where Random has been shown to perform well.
> > * **Random vs. Bonsai-rknn:** We apprehend that the reviewer may have confused Bonsai-rknn with Random. Hence, we now separately show the performance of both in our ablation study (`Fig. 4`). While in Bonsai-rknn, we randomly add computation trees till the distillation budget is exhausted, in Random, we iteratively add random nodes, till their induced subgraph exhausts the budget. Consequently, Random covers more diverse nodes with partial local neighborhood information, while Bonsai-rknn compromises diversity by selecting a smaller node set with complete $L$-hop topology, enabling precise GNN embedding computation. Both represent two distinct mechanisms of random selection. In most cases, Bonsai-rknn achieves a higher accuracy indicating that obtaining full $L$-hop topological information for a smaller set of nodes leads to better results than partial $L$-hop information over a broader set of nodes. We note that random computation tree sampling has thus far not been explored in the graph distillation literature to the best of our knowledge.

---

> > > ### Author Response · Authors · 2024-11-19
> > > **Response to Reviewer n8k1 - Part 3**
> > >
> > > **Q3. From my perspective, selecting datasets using a sampling strategy seems more akin to traditional graph reduction, sparsification, or coarsening methods, rather than directly aligning with the field of graph condensation. Therefore, it’s challenging to accept the significant improvements claimed over random or herding baselines. Could the authors provide an intuitive example to support their approach?**
> > >
> > > **Answer:** Bonsai does not sample trees. It is a deterministic algorithm to select a subset of computation trees that best approximates the embeddings of those computation trees that are filtered out. Sampling can be used in Reverse k-nn computation to approximate tree ranking in exchange of efficiency, but not a necessity. Let us outline the logical progression based on which Bonsai is based.
> > >
> > > 1. **Similar computation trees = similar GNN embeddings:** The embedding of a node is a function of its $L$-hop computation tree. Hence, if two computation trees are similar, their GNN embeddings are also similar regardless of the underlying message-passing architecture (`Hypotheses 1 and Figure 2`).
> > > 2. **Similar embeddings= similar gradients:** When two embeddings are similar, the gradients they generate are similar (`Table 3`). **Core question:** This raises the question: *Given the set of all computation trees, can we select the subset that best approximates the remaining computation trees?*
> > > 3. **Reverse $k$-NN cardinality indicates the approximation power of a tree:** It is due to these observations, we design the reverse $k$-NN based ranking of tree (or node) importances for distillation (also an important distinction from Mirage). Specifically, if tree $T_1$ is among the $k$-NN of tree $T_2$ for a small $k$ (we use $k=5$ for all datasets), this indicates these two trees are similar in WL-embedding. Hence, we seek to include those trees in the distilled set that reside in the $k$-NN of lots of other trees. Consequently, if these trees are selected, their GNN embeddings are also likely similar to their $k$-NN neighbors from the WL space. As a result, they can effectively approximate the GNN embeddings of the filtered-out nodes. Since similar GNN embeddings lead to similar gradients (`Table 3`), we minimize the information lost from nodes that are filtered out.
> > > 4. **Coverage Maximization to identify optimal subset:** The tree selection algorithm begins with the reverse $k$-NN set of each computation tree as input. It then iteratively selects trees based on their *marginal* cardinality - specifically, choosing the tree that appears in the $k$-NN sets of the largest number of yet-uncovered trees. A tree is considered uncovered if none of its $k$-nearest neighbors have been selected for the distilled set. This focus on **marginal contribution naturally promotes diversity**. Consider two similar trees, $T_1$ and $T_2$, both with high reverse $k$-NN cardinality. Due to the transitivity of distance functions, these trees likely share many of the same neighbors in their reverse $k$-NN sets. Consequently, if $T_1$ is selected, $T_2$'s marginal cardinality significantly decreases despite its high initial reverse $k$-NN cardinality, preventing redundant selection of similar trees.
> > >
> > > Random tree selection does not account for either diversity or its approximation power of other trees in the set. Herding, on the other hand, is a cluster the GNN embeddings and opts for the centers. It does not perform the critical ranking of computation trees based on its neighborhood density. Additionally, clusters often have varying diameters, meaning cluster membership alone doesn't guarantee that two trees can effectively approximate each other.
> > >
> > > Finally, we note that our codebase has been released for easy replication of all reported results.
> > >
> > > ----------------
> > >
> > > ### Appeal the reviewer
> > >
> > > In addition to the above clarification, we have now also added data on carbon emissions of all algorithms and Bonsai (Table 9). With our proposed approach, we outperform existing condensation algorithms in accuracy (`Tables 5 and 6`), at least 7 times faster in running time (`Table 7`), and at least 17 times lower in carbon emissions (`Table 9`). More importantly, the proposed distillation algorithm is the only algorithm that is faster than full dataset training with lower carbon emissions; existing algorithms are both slower and have higher carbon emissions, putting into question its practical utility. We hope these results would convince the reviewer on the merits of our work.

---

> > > > ### Author Response · Authors · 2024-11-24
> > > > **Eagerly awaiting feedback before closure of discussion phase.**
> > > >
> > > > Dear Reviewer n8k1,
> > > >
> > > > As we are days away from the closure of the discussion phase, we are eagerly awaiting your feedback on the revisions made. Our rebuttal discusses in details how your feedback has been incorporated. We have also uploaded a revised manuscript that includes all suggested changes such as more detailed ablation study, more clear distinction with Mirage, and several other clarification. In addition, based on suggestion from other reviewers, we have added $3$ more baselines. Overall, our results show that Bonsai is at least 7-times faster than baselines, produces 17-times lower carbon footprint, and more accurate on average across datasets as well as GNN architectures. These benefits come along with the advantage of being model-agnostic. Our codebase has been released for easy replication.
> > > >
> > > > We are keenly awaiting your feedback on the changes made and whether we can do anything more to convince you of the merits of our work.
> > > >
> > > > regards,
> > > >
> > > > Authors

---

> ### Comment · Reviewer_n8k1 · 2024-11-25
>
> Thanks for the author's detailed response. I appreciate your efforts and several last questions and suggestions:
>
> 1. What's the difference between graph distillation and graph condensation? If there's no difference, why do you choose to term your work graph distillation rather than graph condensation, where the latter is much more prevalent in the graph learning community?
>
> 2. Can you further polish Figure 3? The current version looks very sloppy, with the main issues being:  1) It is overly simplistic, conveying very limited information to support your ideas.  2) The overall style of the diagram does not align with the style of other charts in your manuscripts.  3) The input and output on the left suggest that the number of nodes in the diagram seems unchanged; it merely specifies the edges.

---

> > ### Author Response · Authors · 2024-11-25
> > **Incorporating suggestions of Reviewer n8k1**
> >
> > Dear Reviewer n8k1,
> >
> > Thank you for your engagement during the discussion phase. We have carefully addressed your questions below.
> >
> > **Q1. Distillation vs. Condensation:** There is no difference. In fact, we mention this explicitly in line 33 of the first paragraph of our manuscript. We agree that condensation is the more popular keyword in graph condensation space. In the vision community, several papers have used the term distillation instead of condensation, which motivated the seminal GCOND paper (See [1] [2] [3]). **We would be glad to change "distillation" to "condensation" in our title and rest of the manuscript to align more closely with current terminology in the field.**
> >
> > [1] Ondrej Bohdal,Yongxin Yang, and Timothy Hospedales. Flexible dataset distillation: Learn labels instead of images, In 4th Workshop on Meta-Learning (MetaLearn) at NeurIPS 2020.
> >
> > [2] Tongzhou Wang, Jun-Yan Zhu, Antonio Torralba, and Alexei A Efros. Dataset distillation. ArXiv preprint,2018.
> >
> > [3] Dataset Distillation by Matching Training Trajectories. George Cazenavette, Tongzhou Wang, Antonio Torralba, Alexei A. Efros, Jun-Yan Zhu, CVPR 2022.
> >
> > **Q2. Suggested changes in Figure 3:** We have revised Figure 3 based on your suggestions to provide a more accurate representation of our algorithm. Please see the updated manuscript pdf for the revised figure.
> >
> > We look forward to your feedback on whether the updated manuscript addresses your queries comprehensively.
> >
> > regards,
> >
> > Authors

---

> > > ### Author Response · Authors · 2024-11-27
> > > **Keenly awaiting your feedback on the updated figure and clarifications**
> > >
> > > Dear Reviewer n8k1,
> > >
> > > Since we will be unable to update our manuscript after 27th, we are very keen to know if the current version satisfactorily addresses all concerns. We have updated the figure as per your suggestions. We have also clarified that distillation and condensation are synonymous in our context and we would be happy to change "distillation" to "condensation" in our manuscript.
> > >
> > > regards,
> > >
> > > Authors

---

> > > > ### Author Response · Authors · 2024-11-27
> > > > **Eagerly awaiting feedback on the revised manuscript**
> > > >
> > > > Dear Reviewer n8k1,
> > > >
> > > > We apologize if our repeated reminders bother you. However, since the deadline to update the manuscript closes today, we are extremely keen to know if the changes made satisfactorily addresses the concerns raised. We hope, with the new experiments, clarifications and improved presentation, you will feel convinced on the merits of our work. Your support to reconsider the rating based on the updated manuscript will be really valued.
> > > >
> > > > regards.
> > > >
> > > > Authors

---

> > > > > ### Comment · Reviewer_n8k1 · 2024-11-27
> > > > >
> > > > > Thanks for your additional work. All my concerns have been addressed. I will raise my ratings.

---

> > > > > > ### Author Response · Authors · 2024-11-27
> > > > > > **thank you for your suggestions and support**
> > > > > >
> > > > > > Dear Reviewer n8k1,
> > > > > >
> > > > > > thank you for your engagement during the discussion phase. Your suggestions have helped us improve our work and we truly value your support for the revised manuscript.
> > > > > >
> > > > > > regards,
> > > > > >
> > > > > > Authors

---

### Official Review · Reviewer_t6gR · 2024-11-04

**Soundness:** 3
**Presentation:** 3
**Contribution:** 2
**Rating:** 6
**Confidence:** 4

**Summary:**

This paper studies critical limitations in existing graph distillation methods. It introduces a new approach that optimizes the training of GNNs by generating smaller, representative datasets without requiring full training on the original data. The proposed method leverages a computation tree-based approach to create a distilled representation that is efficient, adaptable, and capable of maintaining high accuracy across varied datasets and models. Experimental results have demonstrated its effectiveness and efficiency.

**Strengths:**

1. The proposed gradient-free approach bypasses the need for computationally expensive gradient calculations, resulting in a significantly faster distillation process. This efficiency makes Bonsai highly scalable even for large datasets.
2. This model-agnostic method is interesting and saves efforts in hyperparameter tuning when changing condensation models.
3. It is the first distillation method that retains the original node features and synthesizes graphs with unweighted edges, which more faithfully represent the original graph structure.

**Weaknesses:**

1. The idea of Bonsai is very similar to MIRAGE[1], as both methods select frequent trees. This similarity makes Bonsai appear to be a minor adaptation of MIRAGE. Furthermore, much of the theoretical analysis, such as Graph Isomorphism, is borrowed from MIRAGE. Although these two works focus on different tasks, it's strongly recommended to discuss the differences between Bonsai and MIRAGE in the related work section.
2. This paper claims that "distilling to fully-condensed graph" is a problem for previous work. However, most prior methods include a sparsification step, setting a threshold to obtain a sparse graph. Consequently, the number of edges reported in Table 2 is inaccurate. For correct edge counts, please refer to Table 5 in the GCond paper [2].
3. Although this paper presents comprehensive deductions in the theoretical section, some hypotheses appear to be too strong and lack support. For example, Hypothesis 1 (line 201) and Logical Progression 2 (line 215) may not hold true.
4. In Fig. 2, the authors empirically demonstrate the correlation between GNN embedding and WL embedding. When the threshold is large, almost all node pairs are considered, leading to a low correlation. How does this observation inform the method design?
5. In line 235, **Diversity** is highlighted. Which part of the method addresses this concern?
6. To select the trees with top representativeness, why do the authors choose Reverse K-NN? Would it be possible to simply adopt clustering instead?
7. The experimental settings differ from commonly used ones in the following ways: (a) Different dataset split (i.e., training/validation/test set split) (b) Different metric for compression rate. The authors are suggested to clarify the reasons for choosing a different setting.


### Minor
1. The details of PPR starting from line 332 could be simplified or moved to the Appendix as people are familiar with it.
2. Why is Herding OOT (out of time) for Reddit? In my understanding, this method should be efficient.

### References
[1] Mridul Gupta, Sahil Manchanda, HARIPRASAD KODAMANA, and Sayan Ranu. Mirage: Model-
agnostic graph distillation for graph classification. In ICLR 2024

[2] Wei Jin, Lingxiao Zhao, Shichang Zhang, Yozen Liu, Jiliang Tang, and Neil Shah. Graph condensa-
tion for graph neural networks. In ICLR, 2021.

**Questions:**

Please see the weaknesses

---

> ### Author Response · Authors · 2024-11-19
> **Response to Reviewer t6gR- Part 1**
>
> We appreciate the reviewer’s positive feedback and constructive suggestions. We have carefully considered the comments and incorporated the recommended changes, as well as addressed the concerns raised, as outlined below.
>
> **Q1. The idea of Bonsai is very similar to MIRAGE[1], as both methods select frequent trees. This similarity makes Bonsai appear to be a minor adaptation of MIRAGE. Furthermore, much of the theoretical analysis, such as Graph Isomorphism, is borrowed from MIRAGE. Although these two works focus on different tasks, it's strongly recommended to discuss the differences between Bonsai and MIRAGE in the related work section.**
>
> **Answer:** We have updated the manuscript (`Sec 1.1` and `App A.1`) with a detailed discussion on the novel contributions of Bonsai with respect to Mirage.
>
> The similarities between Mirage and Bonsai end at their shared goal of analyzing computation trees for distillation. Their approaches and capabilities differ fundamentally. Specifically,
> * **Bonsai does not select frequent trees**.
> * **Bonsai does not perform graph isomorphism either.**
> * Bonsai selects a set of trees located in **dense neighborhoods** and **diverse** from each other.
>
> Let us elaborate.
>
> **Summary of Mirage:** Given a graph database, Mirage identifies all unique computation trees of a certain depth in the graph database. Two trees are non-unique if they are isomorphic to each other. Next, each graph is represented as a set of computation trees, on which ferquent itemset mining is performed. Trees in these frequent itemsets form the distilled dataset.
>
> **What breaks in our setting?** Tree isomorphism does not work when each node is attributed with high-dimensional feature vectors since treating feature vectors as labels means two trees are isomorphic if they are topologically identical and the **mapped nodes across the graphs are annotated with identical feature vectors.** Mirage explcitly looks at only graphs where nodes are annotated with a single discrete label (such as atom-types in chemical compounds) and hence tree isomorphism can be performed. In Bonsai, we make no such assumption and hence any algorithm built on tree isomorphism is not feasible.
>
> Bonsai makes the following novel contributions that shares no similarity to Mirage:
>
> * Bonsai embeds computation trees into a feature space using WL-kernel and **ranks** each tree based on the density in its neighborhood (Reverse $k$-NN).
> * Reverse K-NN is expensive ($O(n^2)$). Bonsai proposes an efficient sampling strategy with provable guarantees on the sample_size-approximation_error trade-off.
> * The core idea is to select a subset of trees that are representative of the entire set. Hence, we select trees located in dense neighborhoods and diverse from each other. This ensures that all trees that are not selected in distilled set are likely to have a close neighbor (in the embedding space) in the distilled set. This is achieved through coverage maximization (`Sec 3.2`).
> * We prove coverage maximization is NP-hard, monotonic and submodolar. Hence, greedy selection provides $1-1/e$ approximation guarantee.
> * Sparsification of the distilled dataset is performed through personalized page rank.
>
> **Empirical validation:** To further leave no room for ambiguity, we applied Mirage on Cora and Citeseer, and in both dataset all computation trees were unique leading to no compression.

---

> > ### Author Response · Authors · 2024-11-19
> > **Response to Reviewer t6gR- Part 2**
> >
> > **Q2. This paper claims that "distilling to fully-condensed graph" is a problem for previous work. However, most prior methods include a sparsification step, setting a threshold to obtain a sparse graph. Consequently, the number of edges reported in Table 2 is inaccurate. For correct edge counts, please refer to Table 5 in the GCond paper [2].**
> >
> > **Answer:** We apologize for this inaccuracy and not articulating our point precisely. However, as we will argue below, the broader claim that the distilled graph being larger than the full graph in some cases remains intact. We present the updated discussion from the paper verbatim below. We also present the updated Table 2 with corrected edge counts below supporting this claim.
> >
> > >In a message-passing GNN, the computation cost of each forward pass is $\mathcal{O}(|E|)$, where $E$ denotes the set of edges. Consequently, the computational effectiveness of graph distillation is primarily determined by the reduction in edge count between the original and distilled graphs, rather than node count alone. However, current graph distillation algorithms (see Table 1) quantify the condensation ratio based on the node count. Specifically, given a compression ratio $r\_n$, it synthesizes a weighted, fully-connected dense adjacency matrix for the distilled graph of the size $\frac{|V|}{r\_n}\times \frac{|V|}{r\_n}$, where $V$ denotes the node set. **Some of these algorithms sparsify by removing edges with edge weights below a certain threshold. This threshold is chosen by studying the drop in accuracy at various sparsity levels and choosing a point providing good accuracy-efficiency trade-off. Consequently, the sparsification process itself requires training on the fully connected initial distilled graph.** This design, which is inconsistent with the computation structure of GNNs, can lead to a distilled graph with small reduction in edge count and, in some cases, may even result in a graph with more edges than the original dataset (See Table 2).
> >
> > | Dataset | GCond | GDEM | Full Dataset |
> > |---------|-------|------|--------------|
> > | Cora (5.2%) | **15,074** | **19,600** | 10,556 |
> > | Citeseer (3.6%) | **10,996**| **14,400** | 9,104 |
> > | Pubmed (0.3%) | 3,557 | 3,600 | 88,648 |
> > | Flickr (1%) | 23,556 | 795,664 | 899,756 |
> > | Ogbn-arxiv (0.5%) | 15,110 | 715,716 | 2,315,598 |
> > | Reddit (0.2%) | 5,756 | 216,225 | 23,213,838 |
> >
> > **Table caption:** We present the number of edges in the distilled graphs produced by GCOND and GDEM and compare them to the full dataset. **Bold** cells indicate cases where the distilled graphs have more edges than the full dataset. The indicated node ratios are taken from the values used in GCOND. For GCOND, we report the number of edges after sparsification as reported in their github repository at https://github.com/ChandlerBang/GCond/tree/main/saved_ours.
> >
> > **Q3. Although this paper presents comprehensive deductions in the theoretical section, some hypotheses appear to be too strong and lack support. For example, Hypothesis 1 (line 201) and Logical Progression 2 (line 215) may not hold true.**
> >
> > **Answer:** Thank you for this important feedback. We acknowledge that our original statement was based on intuitive algorithm design principles rather than theoretical guarantees. Following your suggestion, we conducted empirical analysis to examine the relationship between WL-embedding similarities and training gradients.
> >
> > The results lends support to our hypothesis, revealing statistically significant ($p$-value$<0.05$) positive correlations:
> >
> > | Dataset | Correlation | $p$-value |
> > | --- | --- | --- |
> > | Cora | 0.74 | $\approx 0$ |
> > | Citeseer | 0.83 | $\approx 0$ |
> > | Pubmed | 0.38 | 0.02 |
> > | Reddit | 0.42 | $\approx 0$ |
> >
> > **Changes in manuscript:** We have incorporated these empirical findings in `Sec 3` of the revised manuscript.
> >
> > **Q4. In Fig. 2, the authors empirically demonstrate the correlation between GNN embedding and WL embedding. When the threshold is large, almost all node pairs are considered, leading to a low correlation. How does this observation inform the method design?**
> >
> > **Answer:** It is due to this observation, we design the reverse $k$-NN based ranking of node importances for distillation (also an important distinction from Mirage). Specifically, if node $v$ is among the $k$-NN of node $u$ for a small $k$ (we use $k=5$ for all datasets), this indicates these two nodes are similar in WL-embedding. Hence, we seek to include those nodes in the distilled set that reside in the $k$-NN of lots of other nodes. Consequently, if these nodes are selected, their GNN embeddings are also likely similar to their $k$-NN neighbors from the WL space. As a result, they can effectively approximate the GNN embeddings of the filtered-out nodes. Since similar GNN embeddings likely lead to similar gradients (as discussed in Q3 above), we minimize the information lost from nodes that are filtered out.
> >
> > We have now clarified this explicitly in `Sec 3`.

---

> > > ### Author Response · Authors · 2024-11-19
> > > **Response to Reviewer t6gR- Part 3**
> > >
> > > **Q5. In line 235, Diversity is highlighted. Which part of the method addresses this concern?**
> > >
> > > **Answer:** Diversity in our approach is achieved through **coverage maximization (`Sec 3.2`)**, another key distinction from Mirage. The process, detailed in `Algorithm 1`, works as follows:
> > >
> > > The algorithm begins with the reverse $k$-NN set of each computation tree as input. It then iteratively selects trees based on their *marginal* cardinality - specifically, choosing the tree that appears in the $k$-NN sets of the largest number of yet-uncovered trees. A tree is considered uncovered if none of its $k$-nearest neighbors have been selected for the distilled set.
> > >
> > > This focus on **marginal contribution naturally promotes diversity**. Consider two similar trees, $T_1$ and $T_2$, both with high reverse $k$-NN cardinality. Due to the transitivity of distance functions, these trees likely share many of the same neighbors in their reverse $k$-NN sets. Consequently, if $T_1$ is selected, $T_2$'s marginal cardinality significantly decreases despite its high initial reverse $k$-NN cardinality, preventing redundant selection of similar trees.
> > >
> > > We have now updated our manuscript (`Sec 3.2` as well as start of `Sec 3`) to incorporate the above discussion. Thanks for your suggestion.
> > >
> > > **Q6. To select the trees with top representativeness, why do the authors choose Reverse K-NN? Would it be possible to simply adopt clustering instead?**
> > >
> > > **Answer:** Bonsai's effectiveness stems from its explicit **ranking** of computation trees based on their representative power - measured by how many filtered-out trees they can effectively approximate through $k$-NN neighborhood relationships (`Equation 3`).
> > >
> > > Traditional clustering approaches cannot achieve this critical ranking requirement. Additionally, clusters often have varying diameters, meaning cluster membership alone doesn't guarantee that two trees can effectively approximate each other. This limitation is evident in the baseline Herding method, which essentially performs $k$-means clustering on GNN embeddings. Bonsai's superior performance compared to Herding demonstrates the advantages of our reverse $k$-NN approach.
> > >
> > > In general, we believe potential alternatives to reverse $k$-NN would be the literature from space-partitioning algorithms such as locality sensitive hashing. We hope to explore such alternatives in our future works.

---

> > > > ### Author Response · Authors · 2024-11-19
> > > > **Response to Reviewer t6gR- Part 4**
> > > >
> > > > **Q7. The experimental settings differ from commonly used ones in the following ways: (a) Different dataset split (i.e., training/validation/test set split) (b) Different metric for compression rate. The authors are suggested to clarify the reasons for choosing a different setting.**
> > > >
> > > > **Answer:** Indeed this is an important aspect of our study.
> > > >
> > > > * **Different metric for compression:** As discussed in `Section 1.1 (line 64 onwards)`, compression has predominantly been measured by counting nodes in the distilled dataset. However, this metric overlooks several crucial factors. First, the efficiency of a GNN scales linearly with the number of edges, a factor not captured by node-based compression ratios. Additionally, GPU memory consumption depends on both the edge count in the ℓ-hop neighborhood of a node and the number of node attributes. This limitation becomes particularly evident in datasets like Cora and Citeseer (`Table 2`), where the distilled datasets produced by Gcond and GDEM actually contain more edges than the full dataset! This significant detail has remained obscured due to the exclusive focus on node count as the compression metric. A more comprehensive approach would consider the total byte consumption of the distilled dataset, providing a holistic measure that accounts for all these factors.
> > > >
> > > > * **Different dataset split and compression ratios:** The compression ratios established in GCond have become a de facto standard, with subsequent works largely adopting the same approach. However, our analysis reveals several significant issues with this methodology that necessitate a fresh examination.
> > > >
> > > >   * The first concern is the inconsistency in compression ratios across datasets. For instance, while Cora uses ratios of 1.3%, 2.6%, and 5.2%, Reddit employs much smaller ratios of 0.05%, 0.1%, and 0.2%. Notably, there appears to be no documented justification for these widely varying ratios. Our approach **implements uniform compression ratios across all datasets, eliminating potential dataset-specific biases.**
> > > >
> > > >   * The second issue stems from the interaction between data splits and compression ratios. The lack of uniformity in both these aspects makes the compression ratios difficult to interpret meaningfully. As examples, in GCond and GDEM, the **compression ratios of 5.2% and 3.6% in Cora and Citeseer actually translate to using all nodes in the training set** as these percentages match the proportion of nodes in the training data. Thus, in practice, it means no compression with respect to node count, and inflation with respect to edge counts (`Table 2`). Such crucial insights remain hidden due to the inconsistent application of splits and condensation ratios across datasets.
> > > >
> > > > By standardizing both data splits and compression ratios across datasets, we provide more transparent evaluation, clearer understanding of true compression effectiveness, fair comparison across different datasets, and unambiguous measurement of compression relative to original datasets. We believe this revised approach enables more meaningful insights into the effectiveness of different distillation methods.
> > > >
> > > > **Minor Comments:**
> > > >
> > > > **Q8. The details of PPR starting from line 332 could be simplified or moved to the Appendix...**
> > > >
> > > > **Answer:** We acknowledge this feedback. We have moved PPR to appendix.
> > > >
> > > > **Q9. Why is Herding OOT (out of time) for Reddit? In my understanding, this method should be efficient.**
> > > >
> > > > **Answer:** We have added Herding results for Reddit (`Table 5`). Your observation is correct, and it appears to be an Out-of-memory error that ocurred during our execution, perhaps due to multiple workloads. We have throughly verified that the OOT reported for GCSR are indeed OOT. GCSR trains on the full train set 100 times, and hence the high running time is expected.
> > > >
> > > > -------
> > > > ### Appeal to the reviewer
> > > >
> > > > In addition to the clarifications made above, we have also added data on the carbon emissions of the various algorithms (`Table 9`). The data shows that all existing techniques except Bonsai are in fact slower than training on the full train set (`Table 7`) and have higher carbon emissions (`Table 9`). We hope these results would convince the reviewer on the merits of our work.

---

> > > > > ### Author Response · Authors · 2024-11-23
> > > > > **Eagerly awaiting feedback from Reviewer t6gR**
> > > > >
> > > > > Dear Reviewer t6gR,
> > > > >
> > > > > We thank you for your constructive suggestions on our work. Based on your suggestions, our updated manuscript includes:
> > > > > * Detailed differentiation with Mirage
> > > > > * Clarifications and updated discussions to highlight the novelty and importance of reverse $k$-NN, ensuring diversity through coverage maximization, empirically validating the hypotheses of similar embeddings leading to similar gradients, etc.
> > > > > * Justification of our empirical setup and why we believe it's a more fair and transparent setup to evaluate graph distillation algorithms.
> > > > > * New experiments to show Bonsai consumes 17 times lower carbon foot prints, 7 times faster and overall more accurate than existing algorithms.
> > > > >
> > > > > We are keenly awaiting your reaction to the changes made and if there is anything more we can do to convince you of the merits of our work.
> > > > >
> > > > > regards,
> > > > >
> > > > > Authors

---

> > > > > > ### Author Response · Authors · 2024-11-26
> > > > > > **Awaiting your response**
> > > > > >
> > > > > > Dear Reviewer t6gR,
> > > > > >
> > > > > > We really appreciate your detailed feedback and have made changes to our manuscript to address them. We are extremely keen to know if there are any outstanding concerns following our rebuttal. As noted by ICLR, 27th Nov is the last date to make changes to the manuscript. Hence, your engagement within this deadline would be much appreciated.
> > > > > >
> > > > > > regards,
> > > > > >
> > > > > > Authors

---

> > > > > > > ### Comment · Reviewer_t6gR · 2024-11-27
> > > > > > >
> > > > > > > Thank you for the detailed response. I am willing to improve my score and strongly recommend incorporating the feedback into the revision to avoid any misinterpretation.

---

> > > > > > > > ### Author Response · Authors · 2024-11-27
> > > > > > > > **thank you for supporting our revised manuscript**
> > > > > > > >
> > > > > > > > Dear Reviewer t6gR,
> > > > > > > >
> > > > > > > > We appreciate your support for our work. We have already incorporated all of the suggestions made in our revised manuscript. We still have few hours left to update the manuscript, if there are any further suggestions, we will ensure to incorporate those as well.
> > > > > > > >
> > > > > > > > Thank you again for raising the score.
> > > > > > > >
> > > > > > > > regards,
> > > > > > > >
> > > > > > > > Authors

---

### Official Review · Reviewer_YqxY · 2024-11-07

**Soundness:** 3
**Presentation:** 2
**Contribution:** 2
**Rating:** 6
**Confidence:** 3

**Summary:**

This paper proposes a novel graph distillation method empowered by the observation that computation trees form the fundamental processing units of message-passing GNNs. This paper specifically addresses the issue of overly dense edges in graph distillation. Experiments on various datasets verify the effectiveness of the method.

**Strengths:**

1. Compared to previous works, BONSAI is novel.
2. The experimental results look very good, especially regarding the training time.
3. The theoretical analysis is solid.

**Weaknesses:**

1. This paper is not easy to understand.
2. In some cases, BONSAI does not perform the best, such as with citeseer.
3. Regarding table 5, can you provide experimental results for other compression rates?
4. PPR and RkNN involve many parameters, and the ablation study in Fig. 4(b) is insufficient.

**Questions:**

See weakness

---

> ### Author Response · Authors · 2024-11-19
> **Response to Reviewer YqxY- Part 1**
>
> We thank the reviewer for their positive comments on our work as well as the constructive feedback. Please find below our responses to the concerns raised.
>
> **Q1. This paper is not easy to understand.**
>
> **Answer:** We acknowledge this feedback. To improve comprehension of the proposed work, we have now significantly expanded the start of Section 3. In this section, we present a detailed outline of the logical foundation that Bonsai is built upon as well as empirical data supporting this foundation. In general, aim to orient the user towards the high-level design of our algorithm before detailing the individual steps.
>
> We hope this would enhance the readability of our work. We would be happy to incorporate any further suggestions from the reviewer in this regard.
>
> **Q2. In some cases, BONSAI does not perform the best, such as with citeseer.**
>
> **Answer:** While Bonsai may not be optimal in every scenario, our evaluation shows it offers superior performance and consistency across datasets. The evidence is compelling:
>
> -   Bonsai achieves top accuracy in 12 out of 18 test cases - significantly outperforming the next best method, GDEM, which leads in only 2 out of 18 cases.
> -   In the 6 cases where Bonsai isn't the top performer, it ranks second in 3, demonstrating robust reliability.
> -   Most notably, Bonsai delivers this robust performance without requiring full-dataset training, resulting in more than 7-times speed-up (`Table 7`) over the fastest competitor. Furthermore, Bonsai is CPU-bound resulting in and at least 17-times reduced carbon emissions (`Table 9`) than any of the existing distillation algorithms.
>
> This combination of consistent accuracy, reduced computational needs, and lower carbon footprint makes Bonsai unique compared to existing graph distillation works.
>
> **Q3. Regarding table 6, can you provide experimental results for other compression rates?**
>
> **Answer:** We have expanded `Table 6` to include all three compression rates of 0.5%, 1% and 3% in the updated manuscript. The same table is also provided below for each reference. Bonsai outperforms all baselines across all compression rates.
>
> ### Model Accuracy Comparison
>
> | **Dataset**   | **%** | **GNN** | **Random**| **Herding**  | **GCond**  | **GDEM**  | **GCSR** | **Bonsai**  | **Full** |
> |---|--|---|----|---|--|--|---|---|--|
> |  | 0.5   | GAT|41.44±1.73 | 33.80±0.07 | 13.21±1.99| 63.91±5.91†| 15.09±6.19 | **75.42±1.61**   | |
> | | 1|GAT|42.73±1.03 | 46.09±0.86 | 35.24±0.00| 73.49±2.64†|37.60±1.34|**78.67±0.89**|85.70±0.09      |
> | Cora     |3|GAT|60.22±0.67   | 56.75±0.45 | 35.24±0.00  | 75.28±4.86†  | 36.72±0.81  | **80.66±0.80**  |  |
> |  | 0.5   | GIN|49.04±0.50  | 34.39±1.03 | 14.13±6.80 | 63.65±7.11 | 76.05±0.44†  | **85.42±0.74**||
> |  |1|GIN|50.48±0.85|33.80±2.42|33.91±1.23|75.92±4.24†|60.70±4.44|**84.80±0.41**|86.62±0.28      |
> |  |3|GIN|59.52±0.88|36.35±0.59|31.70±4.97|59.59±7.95†|51.62±5.00|**85.42±0.53**||
> |  |0.5   | GAT|42.76±0.35|36.04±0.46|21.47±0.00|69.86±2.28†|21.92±0.76|**68.56±0.57**||
> |  |1|GAT|46.19±1.38|52.07±0.11†|21.47±0.00|23.87±3.05|21.50±0.06|**69.43±0.82**|77.48±0.75      |
> | CiteSeer |3|GAT|61.65±0.51|65.17±0.00†|21.26±0.22|22.90±1.20|21.50±0.06|**69.94±1.15**||
> |  |0.5   | GIN|44.86±0.43|22.97±0.30|21.47±0.00|67.69±3.28†|50.66±1.17|**71.80±0.26**||
> |  |1|GIN|47.90±0.65|39.67±0.82|19.49±1.09|67.64±4.45†|64.74±1.88|**72.16±0.60**|75.45±0.23      |
> |  |3|GIN|61.83±0.68|60.48±0.26†|18.65±2.56|48.65±8.17|59.95±9.07|**70.51±0.54**||
> |  |0.5   | GAT|77.73±0.12|75.44±0.02|37.49±4.01|80.06±1.16†|38.29±8.13|**85.66±0.38**||
> |  |1|GAT|78.85±0.09|76.64±0.02|41.55±3.18|80.75±0.47†|40.47±0.00|**85.88±0.28**|86.33±0.08      |
> | PubMed   |3|GAT|82.84±0.11†|78.48±0.03|37.77±3.61|65.08±9.53|40.27±0.20|**85.62±0.36**||
> |  |0.5   | GIN|77.45±0.14|48.48±1.33|30.91±4.57|78.78±0.91†|36.88±12.06|**84.32±0.33**||
> |  |1|GIN|78.43±0.22|62.22±0.13|32.84±6.27|78.72±0.95†|33.75±5.58|**85.57±0.26**|84.66±0.05      |
> |  |3|GIN|80.56±0.17|45.40±0.46|36.11±3.47|81.08±0.99†|32.01±6.77|**85.66±0.23**||
> |  |0.5   | GAT|43.64±0.99†|36.50±13.22|40.24±3.20|25.43±10.37|28.03±6.60|**48.22±3.60**||
> |  |1|GAT|43.56±1.06†|36.34±1.14|40.85±1.08|18.44±9.42|OOT|**45.62±1.85**|51.42±0.07      |
> | Flickr   |3|GAT|45.71±1.87†|42.70±1.17|41.51±9.81|25.83±11.39|OOT|**47.80±2.06**||
> |  |0.5   | GIN|42.67±0.83†  | 39.98±7.21|13.65±7.54|14.10±5.68|5.92±1.01|**44.97±2.23**||
> |  |1|GIN|42.90±0.76† | 41.87±4.52|16.65±6.55|19.44±9.68|OOT|**44.90±0.88**|45.37±0.57 |
> |  |3|GIN|19.63±4.21  | 43.72±3.26†|24.25±14.43|20.97±6.64|OOT|**45.04±1.94**  |  |
> Notes:
> - Bold numbers (**) indicate the best performance
> - † indicates the second-best performance
> - OOT indicates "Out of Time"

---

> ### Author Response · Authors · 2024-11-19
> **Response to Reviewer YqxY- Part 2**
>
> **Q4a. PPR and RkNN involve many parameters**
>
> **Answer:** This appears to be a misunderstanding. Bonsai requires only four parameters- far less most other the baselines. The table below presents the parameters required in the config files for each of the distillation algorithms. We re-emphasize some important points with respect to Bonsai that underscores its robustness to parameters.
>
> * As we noted in our introduction, all baselines, except GDEM, mimic gradients on the full train set, and hence requires knowledge of all parameters that will be used while training. In addition, they have some distillation-specific parameters. This design significantly inflates their parameter sets as evident from the table below (in addition to making them order of magnitudes slower).
>
> * Finally, we emphasize that we **did not tune parameters** for each dataset, GNN architecture or distillation ratio--the same values were used across all experiments to showcase our robustness. In contrast, the parameters for all baselines are specific to each combination of dataset-GNN architecture-distillation ratio. While the authors have not pointed out how the optimal values were arrived at, it appears grid search is the only possible methodology, making their deployment in practical settings challenging. This behavior makes Bonsai easier to deploy in production settings on unseen datasets.
>
> * Even when we vary the parameters of Bonsai (`Fig 7`), they hardly affect the quality, demonstrating its robustness and why dataset or compression specific tuning is not necessary.
>
> Bonsai|GCond|GDEM|GCSR|GC-SNTK|GEOM|EXGC
> -----|------|-----|------|-----------------|-----|--
> $k$ in RkNN|Gradient matching parameter: Number of inner iterations|Number of largest eigenvalues to match|Gradient matching parameter: Number of inner iterations|$K$: Number of neighborhood aggregation rounds|$U$: Initial upper limit of expanding window|Gradient matching parameter: Number of inner iterations
> Sample size $z$ in RkNN|Gradient matching parameter: Number of outer iterations|Number of smallest eigenvalues to match|Gradient matching parameter: Number of outer iterations|$L$: Number of iterations in SNTK|$U'$: Upper bound of expanding window|Gradient matching parameter: Number of outer iterations
> $\theta$ in PPR to find knee point|Sparsification parameter|$k$: Number of eigenvectors|Sparsification parameter|$\lambda$: Regularization parameter tuned $10^{-6}$ to $10^6$|$\zeta$: Number of epochs for training in buffer phase|Sparsification parameter
> |Number of layers in WL kernel|Regularization parameter $\alpha$|$\tau_1$|Regularization parameter $\alpha$||Scheduler: Pacing functions (, root, geometric)|Learning rate
> ||Learning rate|$\tau_2$|Learning rate||$lr\_{feat}$: Learning rate for condensed graph features|Dropout
> ||Dropout|Learning rate for eigenvectors|Dropout||$lr\_y$: Learning rate for soft labels (0 for hard labels)|Pruning parameter
> ||Weight decay|Learning rate for features|Weight decay||$p$: Number of training steps for expert GNNs|Mining parameter
> ||Number of epochs|Number of epochs|Number of epochs||$q$: Number of checkpoints for student GNNs|Circulation parameter
> ||Early-stopping patience parameter|Loss weight: $\alpha$|Early-stopping patience parameter||$\alpha$: Weight for knowledge embedding loss term|Number of epochs
> |||Loss weight: $\beta$|Regularization parameter $\beta$ (Eq. 11)|||Regularization parameter $\alpha$
> |||Loss weight: $\gamma$|Number of experts|||Early-stopping patience parameter
> ||||$\tau$ (Eq. 17)|||Weight decay
> ||||$\gamma$ (Eq. 18)
>
> **Q4b. The ablation study in Fig. 4(b) is insufficient.**
>
> **Answer:** Bonsai operates through two main mechanisms: **(i)** exemplar tree selection via reverse $k$-nearest neighbors and **(ii)** sparsification using personalized PageRank (PPR).
>
> * We have  expanded our ablation study to study the impact of each of these components across **all datasets on all compression ratios** (`Fig 4`).
>
> * We present detailed analyses in `Appendix B.4.2` examining how different values of $k$ and sample sizes in reverse $k$-NN influence both model accuracy and computational efficiency. These experiments demonstrate Bonsai's parameter robustness and show that the method performs well without requiring dataset-specific parameter tuning.
>
> Should the reviewer have further suggestions for the ablation study, we would be happy to incorporate them.

---

> > ### Author Response · Authors · 2024-11-23
> > **Eagerly waiting for feedback on revisions made**
> >
> > Dear Reviewer YqxY,
> >
> > We have provided detailed explanations and additional experiments to address your concerns. We have also uploaded a revised manuscript with all new additions clearly highlighted in blue font. Since the discussion phase will close soon, we are keen to get your reaction to the changes made.
> >
> > regards,
> >
> > Authors

---

> > > ### Comment · Reviewer_YqxY · 2024-11-25
> > >
> > > Thank you for your responses. I appreciate your effort and I raise the score.

---

> > > > ### Author Response · Authors · 2024-11-25
> > > >
> > > > We appreciate the reviewer's positive assessment of our revised manuscript and the subsequent increase in score.
> > > >
> > > > regards,
> > > >
> > > > Authors

---

### Author Response · Authors · 2024-11-19
**Cover letter**

We thank the reviewers for their insights and constructive suggestions. A comprehensive point-by-point response to the reviewers' comments is presented below. **We have updated the main manuscript** to address these comments. The changes made in the manuscript are highlighted in **blue** font. The major additional changes are listed below.

- **Additional experiments:** We have incorporated all of the additional experiments requested by the reviewers spanning
    - Three additional baselines of EXGC, GCSNTK and GEOM
    - Expanded ablation study in Fig. 4
    - Additional data on carbon emissions
    - Empirical analysis that highlights relationship between WL embedding similarities and training gradients

- **Presentation:** We have addressed several presentation-related inquiries, such as highlighting differences between Mirage and Bonsai, moving auxiliary details to appendix, and adding clarifications wherever appropriate.

We hope these revisions will satisfactorily address the concerns raised by the reviewers and elevate the overall quality of our work. We remain open to any further suggestions.

---

> ### Author Response · Authors · 2024-11-24
> **Request for rebuttal feedback - Discussion closing soon**
>
> Dear Reviewers YqxY, t6gR, and n8k1,
>
> Thank you for your valuable feedback on our submission. We submitted our detailed rebuttal and revised manuscript on November 19th, addressing your comments and suggestions.
>
> As the discussion phase closes in two days, we would greatly appreciate your thoughts on our revisions. Early feedback would allow us time to address any remaining concerns you may have. We're pleased to note that Reviewer q9ty has reviewed our changes and responded positively, increasing their rating to 6.
>
> Thank you for your time and consideration.
>
> Best regards,
>
> Authors

---

### Comment · Area_Chair_GC58 · 2024-11-28

I would like to encourage the reviewers to engage with the author's replies if they have not already done so. At the very least, please
acknowledge that you have read the rebuttal.

---

### Meta-Review · Area_Chair_GC58 · 2024-12-19

**Metareview:**

This authors proposes a gradient-free graph distillation/condensation method based on the observation that  similar computation trees imply similar embeddings, which in turn imply similar gradients. The method is model-agnostic, retains the original node features, and scalable. The reviewers were convinced by both the idea and the experimental results. The reviewers also appreciated the theoretical analysis (simple but interesting).

**Additional Comments On Reviewer Discussion:**

Several reviewers asked for the difference between the proposed method Bonsai and Mirage. The authors clarified that the main difference is that Bonsai selects diverse trees in dense neighborhoods, while Mirage which selects frequent trees. They were able to convincingly address this and other concerns and subsequently, all four reviewers increased their score from 5 to 6.

---

### Decision · Program_Chairs · 2025-01-22

Accept (Poster)